# Variations in Essential Oil Composition and Chemotype Patterns of Wild Thyme (*Thymus*) Species in the Natural Habitats of Hungary

**Zsuzsanna Pluhár** [1,*] , **Róbert Kun** [2], **Judit Cservenka** [3], **Éva Neumayer** [4], **Szilvia Tavaszi-Sárosi** [1] , **Péter Radácsi** [1] and **Beáta Gosztola** [1]

1 Department of Medicinal and Aromatic Plants, Institute of Horticultural Science, Hungarian University of Agriculture and Life Sciences, H-1118 Budapest, Hungary; tavaszi-sarosi.szilvia@uni-mate.hu (S.T.-S.); radacsi.peter@uni-mate.hu (P.R.); gosztola.beata@uni-mate.hu (B.G.)
2 Department of Nature Conservation and Landscape Management, Hungarian University of Agriculture and Life Sciences, H-2100 Gödöllő, Hungary; rbert.kun@gmail.com
3 Balaton Uplands National Park Directorate, H-8299 Csopak, Hungary; cservenkajudit@bfnp.hu
4 Magosfa Foundation, H-2600 Vác, Hungary; eneum@magosfa.hu
* Correspondence: pluhar.zsuzsanna@uni-mate.hu

**Abstract:** A comprehensive study was conducted on the diversity and characteristics of five *Thymus* species native to Hungary, concerning frequency of occurrence, habitat preferences, essential oil content of the dried flowering shoots, and chemotype patterns determined by GC/MS. Our main aims were to provide an overview of the essential oil diversity of thyme resources and select the best genotypes with potential for cultivation and utilization. Based on the results obtained in 74 populations of 63 localities belonging to 15 regions of the Transdanubian and Northern Hungarian Mountains, considerable essential oil diversity was found. *Thymus pannonicus* (TPA), of generalist character, was proven to be the most frequent species (38 populations), while *T. serpyllum* (TSE) occurred only in two habitats. High average amounts of essential oil (EO) were shown for *T. pannonicus* (0.46 mL/100 g DW), *T. pulegioides* (TPU: 0.47 mL/100 g DW), and *T. serpyllum* (0.59 mL/100 g DW), while low EO accumulating ability was detected in *T. glabrescens* (TGL: 0.26 mL/100 g DW) and in *T. praecox* (TPR: 0.10 mL/100 g DW). In general, the thymol chemotype was the most frequent (34 populations), found together with the related molecules (p-cymene: 26; γ-terpinene: 15), while numerous other monoterpenes (M: geraniol: 12, linalool: 7) or sesquiterpenes (S: germacrene D: 25, β-caryophyllene: 21) were dominant, as well as combined (MS) chemotypes, which were also described in the Eos of *Thymus* species in Hungary. Our findings confirmed that *T. pannonicus* shows potential for cultivation with homogenous drug quality, adequate amounts of essential oil, and stability in EO composition. Data from original habitats also supports its high tolerance and adaptability to diverse environmental conditions, which is advantageous when facing climate change and extremities.

**Keywords:** *Thymus pannonicus*; *Thymus glabrescens*; *Thymus pulegioides*; *Thymus praecox*; *Thymus serpyllum*; *Serpylli herba*; chemical diversity; thymol; monoterpene; sesquiterpene





## 1. Introduction

The genus *Thymus* (*Lamiaceae*, *Nepetoideae*) involves diverse essential oil-bearing species with high intraspecific variability [1]. Besides garden thyme (*Thymus vulgaris* L.), further wild thyme species attract regional or broader interest, where flowering shoots are collected and used for various therapeutic purposes or applied as ornamental plants in gardens [2]. Essential oils and extracts of wild thyme species (*T. sepyllum* L. s. l.) have a long history and relevance in therapy. Active substances of expectorant and spasmolytic preparations have been used in European traditional medicine for centuries [3]. Recent findings have verified that essential oil and polyphenol-rich extracts of *Thymus* species collected in natural habitats

show significant antioxidant potential as well as antibacterial and antifungal activity against various foodborne and human pathogens, respectively. Moreover, the cytotoxic activity of *T. serpyllum* extract and the antitumor potential of the essential oil have also been proven [4–6]. Thus, wild *Thymus* species provide high-quality raw materials that can be applied in various formulations in self-medication as well as in the pharmaceutical, food, cosmetic, and chemical industries [3,6].

Dried flowering shoots of wild thyme (*Serpylli herba*) are listed in the Pharmacopoea Europaea, with a specification for the essential oil (EO) level (min. 3 mL/kg DW) [7]. In addition, bitter compounds, tannins, flavonoids, and phenolic acids have also been determined in the dried aerial parts of wild thyme [3,8]. Essential oil (*Serpylli aetheroleum*) is produced by distillation of collected flowering shoots of various native species and occurs in different parts of Europe, with two gene centers found in Turkey and in the Iberian Peninsula [9]. Monographs [3] refer to the collected plant material as *Thymus serpyllum* s. l. (sensu lato: in a broader sense), so the main terpenoid compound of the EO can also be different (p-cymene, carvacrol, thymol, linalool, or borneol) in the dried aerial parts of wild thyme plants collected. However, it is to be considered that the application of the drug in phytotherapy is attributed to the phenolic monoterpene molecules thymol and/or carvacrol [6].

According to the current taxonomic concept of Jalas (1972) [10], *Thymus* species are considered collective taxa, '*species aggregata*', involving 2–3 microspecies each. Wild thyme taxa with Eurasian distribution, belonging to the Section *Serpyllum* of 120 chamaephyte species, were divided into 6 subsections and possess the highest chromosomal variability within the genus [9]. The Hungarian *Thymus* species can be classified into the following four subsections:

- Subsect. *Isolepides*: *Thymus glabrescens* Willd.—common thyme; *Thymus pannonicus* All.—Pannonian/Large/Hungarian thyme.
- Subsect. *Alternanthes*: *Thymus pulegioides* L.-broad-leaved/mountain thyme.
- Subsect. *Pseudomarginati*: *Thymus praecox* Opiz—creeping thyme.
- Subsect. *Serpyllum*: *Thymus serpyllum* L.-wild/creeping thyme [9].

Previous studies have reported a significant level of essential oil polymorphism concerning our five indigenous *Thymus* species from various habitats in Western Europe, the Nordic and Baltic countries, the Iberian and Balkan Peninsulas, Turkey, Italy, Iran, Slovakia, and Bulgaria, as well as from Ukraine. However, only sporadic data were available about the essential oil compositions of the drug collected from Hungarian wild thyme populations [11–23].

Based on the relevant literature data, considerable essential oil polymorphism could be expected for Hungary, owing to the quite variable habitat conditions, diverse plant communities, and the existence of five collective species and a few hybrids.

The main aim of our studies was to provide a general overview concerning the occurrence, environmental preference, essential oil properties, and chemotype patterns of the populations of five *Thymus* spp. occurring among diverse habitat conditions in the Hungarian Mountain Range. In addition, a general classification of EO chemotypes, their distribution, and frequency in Hungary are also discussed, along with outlines of future applications and cultivation prospects. Our further purpose was to point out the most valuable resources available in the natural habitats and select the high-yielding genotypes with excellent drug quality. With our findings, our aim was to support specific data on proper wild crafting practices, breeding, growing, and applications, as well as grounding the basis of the gene reservation of native Thymus taxa found in various habitats in Hungary.

## 2. Materials and Methods

*Study areas:* Samples were collected from 74 populations of 5 native *Thymus* species (Figure 1) found on different substrata of 63 natural habitats belonging to 15 regions of Hungary, as summarized in Table 1 and in Figure 2, respectively. Data obtained by

species during our studies conducted in the past twenty years (2000–2020) have partly been published [19–21]; however, numerous new findings are also included in this paper to achieve a more complete overview. Fifteen new localities (4_14; 6_20; 8A_23; 8A_28; 8B_34; 8B_35; 9_36; 9_38; 9_39; 14_52; 14_53; 15B_59; 15B_60; 15C_61; 15C_63) with seventeen thyme populations (6_20: a and b) were explored and examined in the last period (unpublished data).

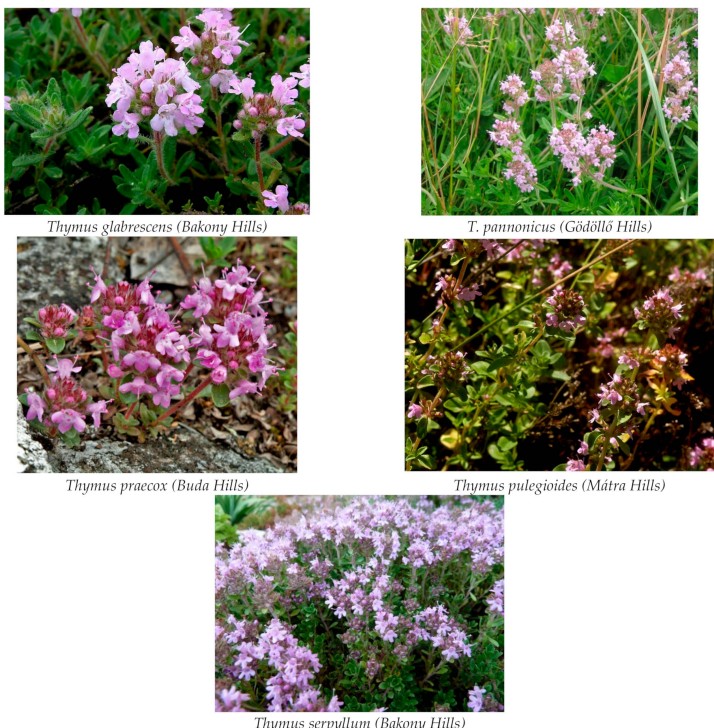

*Thymus glabrescens (Bakony Hills)*

*T. pannonicus (Gödöllő Hills)*

*Thymus praecox (Buda Hills)*

*Thymus pulegioides (Mátra Hills)*

*Thymus serpyllum (Bakony Hills)*

**Figure 1.** *Thymus* species found in natural plant communities in Hungary (Photos by Zs. Pluhár).

**Table 1.** Summary of the geographical locations where populations of *Thymus* species were studied in Hungary with base rocks and soil types (2000–2020).

| Region | Location of Model Area | Site Code Region_Location ** | Species * Found | Base Rock Type | Soil Type |
|--------|------------------------|------------------------------|-----------------|----------------|-----------|
| 1 Mecsek Hills | 1. Kis-Tubes Hill | 1_1 | TPR | gray limestone | black rendzina |
| 2 Somogy Hills | 2. Nagybajom, pasture | 2_2 | TSE | acidic sand | humified sand |
| | 3. Köröshegy, loess hill | 2_3 | TPA | sandy loess | humified sand |
| 3 Balaton Uplands | 4. Várvölgy, Keszthely Hills | 3_4 | TGL | calciferous sand | humified sand |
| | 5. Balatongyörök, Keszthely Hills | 3_5 | TPA | dolomite | bare soil |
| | 6. Zalaszántó, Pap meadows | 3_6 | TPU | brookside silt | meadow soil |
| | 7. Tapolca Basin, Tapolca hillside | 3_7 | TPA | Dachstein limestone | black rendzina |
| | 8. Szentbékkálla, Rock Hill | 3_8a | TPU | Pannonian sandstone | bare soil |
| | | 3_8b | TGL | | |
| | 9. Balatonfüred, Tamás Hill | 3_9 | TPR | dolomite | black rendzina |
| | 10. Balatonfüred, Koloska Valley | 3_10 | TPA | loess | humus carbonate soil |
| | 11. Balatonalmádi, Megye Hill | 3_11 | TPA | loess | bare soil |
| 4 Bakony Hills | 12. Fenyőfő, Pasture Lane | 4_12a | TPA | calciferous sand | humified sand |
| | | 4_12b | TSE | | |
| | 13. Csesznek, Castle Hill | 4_13 | TGL | Dachstein limestone | black rendzina |
| | 14. Szőc, Pasture | 4_14 | TPA | Dachstein limestone | bare soil |
| | 15. Várpalota, Great Meadows | 4_15a | TPR | dolomite | bare soil |
| | | 4_15b | TGL | | |
| | 16. Öskü, Péti Hill | 4_16 | TPA | dolomite | bare soil |
| | 17. Litér, Mogyorós Hill | 4_17 | TPA | dolomite | bare soil |

**Table 1.** *Cont.*

| Region | Location of Model Area | Site Code Region_Location ** | Species * Found | Base Rock Type | Soil Type |
|---|---|---|---|---|---|
| 5 Velence Hills | 18. Pákozd, Moveable Rocks | 5_18 | TGL | granite | bare soil |
| 6 Vértes Hills | 19. Várgesztes, Som Hill | 6_19 | TGL | Dachstein limestone | black rendzina |
| | 20. Csákberény, pasture | 6_20a | TPA | dolomite | black rendzina |
| | | 6_20b | TGL | | |
| 7 Gerecse Hills | 21. Tardosbánya, rock mine plateau | 7_21a | TPA | Dachstein limestone | black rendzina |
| | | 7_21b | TGL | | |
| 8A Buda Hills | 22. Budaörs, Csíki Hills, Odvas Hill | 8A_22a | TPA | dolomite | black rendzina |
| | | 8A_22b | TPR | | |
| | 23. Budapest, Kálvária Hill | 8A_23 | TGL | Dachstein limestone | brown forest soil |
| | 24. Budapest, Újlaki Hill | 8A_24a | TPA | Dachstein limestone | black rendzina |
| | | 8A_24b | TPR | | |
| | 25. Budapest, Vörös-Kővár Hill | 8A_25 | TPA | sandstone of Hárs Hill | black rendzina |
| | 26. Budapest, Homok Hill | 8A_26a | TPA | dolomite | black rendzina |
| | | 8A_26b | TPR | | |
| | 27. Nagykovácsi, Nagy-Szénás Hill | 8A_27a | TPA | dolomite, loess | black rendzina |
| | | 8A_27b | TPR | | |
| | 28. Nagykovácsi, Dog Hill | 8A_28 | TGL | Dachstein limestone | black rendzina |
| | 29. Budapest, Sas Hill | 8A_29 | TPR | dolomite | black rendzina |
| | 30. Érd, Tétény Hills, | 8A_30 | TGL | Sarmathian limestone | black rendzina |
| | 31. Diósd, Tétény Hills | 8A_31 | TPR | Sarmathian limestone | black rendzina |
| 8B Pilis Hills | 32. Dorog, Strázsa Hill | 8B_32 | TGL | calciferous sand | humified sand |
| | 33. Dorog, Park | 8B_33 | TPA | calciferous sand | bare soil |
| | 34. Pilisszentiván, Fehér Hill | 8B_34 | TGL | dolomite | black rendzina |
| | 35. Pilisszántó, Pilis Hill | 8B_35 | TGL | dolomite | bare soil |
| 9 Visegrádi Hills | 36. Szentendre, Dobos Hill | 9_36 | TPA | dolomite | black rendzina |
| | 37. Dömös, Vadálló Cliffs | 9_37 | TPU | andesite | bare soil |
| | 38. Visegrád1, Nagy-Villám | 9_38 | TPA | andesite | black rendzin |
| | 39. Visegrád2, Mogyoró Hill | 9_39 | TPA | andesite | bare soil |
| 10 Börzsöny Hills | 40. Szent Mihály Hill | 10_40 | TPU | andesite | erubase |
| 11 Gödöllő Hills | 41. Veresegyház | 11_41 | TPA | calciferous sand | brown forest soil |
| | 42. Szada | 11_42 | TPA | calciferous sand | brown forest soil |
| | 43. Zsófialiget | 11_43 | TPA | railside soil | bare soil |
| | 44. Ceglédbercel, loess valley | 11_44 | TPA | loess | humified sand |
| | 45. Ceglédbercel, public park | 11_45 | TPA | calciferous sand | humified sand |
| 12 Medves Hills | 46. Salgótarján, Salgó Hill | 12_46 | TGL | basalt | erubase |
| 13 Mátra Hills | 47. Pásztó: Köves Cliff | 13_47 | TGL | andesite | bare soil |
| | 48. Mátrakeresztes: Great Meadows | 13_48a113_48a2 | TPU-L TPU-T | andesite | meadow soil |
| | 49. Sirok, Castle Hill | 13_49 | TPA | rhyolite tuff | bare soil |
| 14 Bükk Hills | 50. Szarvaskő Hilltop | 14_50 | TGL | mudstone | bare soil |
| | 51. Cserépváralja, rhyolit tuff cones | 14_51 | TPA | rhyolite tuff | bare soil |
| | 52. Noszvaj | 14_52 | TPA | mudstone | bare soil |
| | 53. Bogács | 14_53 | TPA | mudstone | black rendzina |
| | 54. Mónosbél, Szappanos Hill | 14_54 | TPA | mudstone | brown rendzina |
| 15A Zemplén Hills | 55. Regéc, meadow | 15A_55 | TPU | rhyolite | bare soil |
| | 56. Regéc, hayland | 15A_56a | TPA | andesite | bare soil |
| | | 15A_56b | TPU | | |
| | 57. Bózsva, Volcanic Cliff | 15A_57 | TPA | rhyolite | bare soil |
| | 58. Vágáshuta, pasture | 15A_58a | TPA | andesite | bare soil |
| | | 15A_58b | TPU | | |
| 15B Aggtelek Hills | 59. Aggtelek, pasture | 15B_59 | TPA | Dachstein limestone | bare soil |
| | 60. Jósvafő, Red Lake, meadows | 15B_60 | TPA | Dachstein limestone | black rendzina |
| 15C Cserehát Hills | 61. Szendrőlád, Szendrő Hills | 15C_61 | TPA | Dachstein limestone | brown forest soil |
| | 62. Rakaca, Szendrő Hills | 15C_62 | TPA | marble | bare soil |
| | 63. Sajógalgóc, Putnok Hills | 15C_63 | TPA | mudstone | bare soil |

Legends: * Abbreviations of species names: TGL: *T. glabrescens*; TPA: *Thymus pannonicus*; TPR: *T. praecox*; TPU: *T. pulegioides*; TSE: *T. serpyllum*; ** a, b marks indicate different species belonging to the same location.

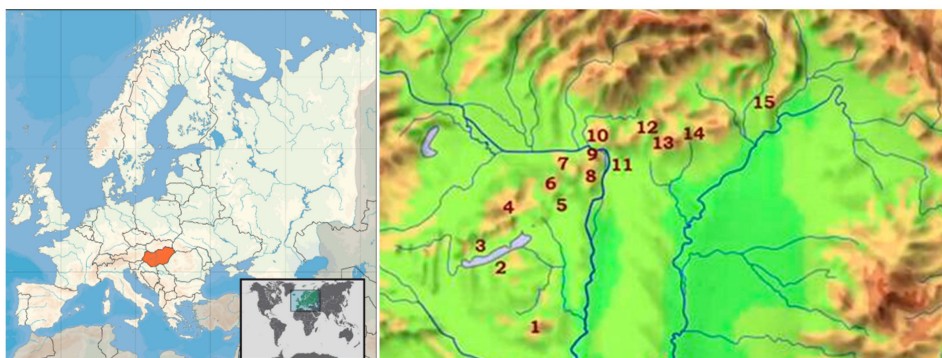

**Figure 2.** Location of Hungary in Europe (https://en.wikibooks.org/wiki/Wikijunior:Europe/Hungary) (accessed on 18 January 2024) and the main geographical regions involving habitats of *Thymus* spp. Populations surveyed in Hungary (2000–2020) (see legends and further details in Table 1).

A wide range of *Thymus* habitats with diverse parent rocks, soil types, and climatic conditions were designated as model areas based on floristic and coenological data recorded by botanists about the occurrence of different thyme populations. Field trips were timed to the flowering period of the species when comprehensive surveys were performed: the exact identification of species and microtaxa, sampling of flowering shoots and soils, habitat description, and preparation of herbarium specimens (Table 1).

*Plant material:* Identification of species was based on the relevant literature, reference book [24], and field data (morphological and coenological traits), as well as on accurate reviews of herbarium specimens. In general, approximately 25–30 g of fresh flowering shoots in *Thymus* populations with 3 to 6 replicates were collected per site, and essential oil data from 2 subsequent years were also evaluated in order to describe the true chemotypes of the localities. Plant samples were dried naturally under indoor conditions and shaded, where room temperature ranged between 15 and 25 °C, directly after collection into marked paper sacks, for approx. 14 days.

When evaluating the EO quality obtained from wild thyme species in our studies, the broader interpretations of *Thymus serpyllum* (s. l.) and *Serpylli herba* (s. l.) were considered, involving all the 5 taxa belonging to the Sect. *Serpyllum* in the genus *Thymus*. Voucher specimens were deposited in the Herbarium of the Department of Medicinal and Aromatic Plants, Buda Campus, Hungarian University of Agriculture and Life Sciences, Budapest.

*Isolation of essential oil*: Dried plant material was hydro-distilled using a Clevenger-type apparatus for 2 h, according to the pharmacopoeial standard, in 3–6 repetitions per population. The essential oil (EO) samples were stored in sealed vials under refrigeration prior to analysis. The EO content of each sample was expressed in ml/100 g on a dried weight basis (DW) and compared with the relevant standard for *Serpylli herba* of Ph. Eur. [7], which specified the minimum level of EO as 3 mL/kg.

*Gas chromatography*: A new analytical method was developed for identifying the exact percentage composition of the essential oil samples. A GC-FID analysis was carried out using an Agilent Technologies 6890N GC System (Santa Clara, California, USA) instrument equipped with an HP-5 (5% phenyl methyl siloxane) capillary column (30 m × d = 350 μm, film thickness: 0.25 μm), programmed as follows: initial temperature of 50 °C (10 min), from 50 to 150 °C at a rate of 4 °C min$^{-1}$; from 150 to 220 °C at a rate of 12 °C min$^{-1}$ and 220 °C (10 min). Carrier gas: helium (constant flow rate of 0.5 mL min$^{-1}$), injector and detector temperatures: 250 °C, split ratio: 22.6:1. Injected quantity: 0.2 μL. The percentage composition of the essential oil was computed from the GC peak areas.

*Gas chromatography–mass spectrometry (GC–MS) analyses* were carried out using an Agilent Technologies 6890N GC System instrument equipped with an HP-5 (5% phenyl methyl siloxane) capillary column (30 m × d = 350 μm, film thickness: 0.25 μm) and connected to an Agilent Technologies MS 5975 inert mass selective detector. The temperature program was the following: initial temperature of 60 °C, then by a rate of 3 °C/min up to 240 °C;

the final temperature was maintained for 5 min. The carrier gas was helium (1 mL min$^{-1}$); injector and detector temperatures were 250 °C. Split ratio: 30:1. Injected quantity: 0.2 μL (1%, solvent: *n*-hexane). Ionization energy was 70 eV. The MS (mass spectra) were recorded in full scan mode, which revealed the total ion current (TIC) chromatograms (mass range m/z 50–550 uma). The compounds were identified by linear retention indices, which were calculated using the generalized equation of Van Den Dool and Kratz [25], literature data, and by matching their recorded mass spectra with those in mass spectral library references (NIST MS Search 2.0 library, Wiley 275, John Wiley & Sons, Inc., Hoboken, NJ, USA), as well as with a home-made database [26]. The calculations of the retention indices were made on a homologous series of alkanes. Relative percentages (%) of compounds were presented in tables and figures in their order of elution in the column.

*Chemotype determination*: In general, chemotypes were characterized on the basis of the main essential oil compounds with a relative percentage of over 10%; however, in exceptional cases, further significant compounds of slightly lower GC% (over 8%) were also indicated. In addition, chemotypes were grouped by the chemical structure of the main compounds into monoterpene (M), sesquiterpene (S), or mixed (MS) classes.

*Statistical analysis*: Data originating from 3 to 6 repetitions of collected samples by populations were involved in statistical analyses. A one-way ANOVA was applied to show significant differences among *Thymus* species concerning essential oil-producing ability as well as to evaluate the effect of habitat conditions provided by regions included in the study. The IBM SPSS Statistics 29.0 software package was used for univariate analysis, where homogeneous groups were separated using the Tukey HSD or Duncan post hoc test, and the mean difference was significant at the $p < 0.05$ level. As a multivariate method, a hierarchical cluster analysis was performed on the basis of the percentage composition of major compounds in individual volatile samples (with compounds over 8%). A dendrogram was obtained using complete linkage with Euclidean distances using the TIBCO Statistica™ 14.0.0 (TIBCO Software Inc., Palo Alto, CA, USA) software package.

## 3. Results

### 3.1. Frequency of Occurrence of Thymus Species

1.  The occurrence of native *Thymus* populations was various, depending on their ecological tolerance, habitat preferences, and social behavior types. Among 74 populations of 5 species surveyed in the Hungarian Mountain Range, *T. pannonicus* (TPA) was found with the highest frequency (38 populations, 51.35%), followed by *T. glabrescens* (TGL) (17 populations, 22.98%) (Tables 1 and 2). Both species were explored at new sites in the last few years, which also verifies their broad ecological tolerance and generalist character (Tables 1 and 2).

**Table 2.** Frequency of occurrence (number, %) of *Thymus* populations surveyed in Hungary and classification of Thymus species according to the social behavior types, according to Borhidi, 1995 [27].

| Species Name | Abbreviation | Overall | | New | | Social Behavior Type |
|---|---|---|---|---|---|---|
| | | No | % | No | % | |
| *Thymus pannonicus* All. | TPA | 38 | 51.35 | 12 | 16.21 | generalist |
| *T. glabrescens* Willd. | TGL | 17 | 22.98 | 5 | 6.76 | generalist |
| *T. praecox* Opiz | TPR | 8 | 10.81 | - | - | specialist |
| *T. pulegioides* L. | TPU | 9 | 12.16 | - | - | generalist |
| *T. serpyllum* L | TSE | 2 | 2.70 | - | - | natural competitor |
| | Total | 74 | 100 | 17 | 22.87 | |

2.  Populations of *T. pannonicus* were found on diverse parent rocks, mainly on limestone (e.g., Buda Hills, Tapolca Basin), dolomite (e.g., Bakony Hills, Vértes Hills), and loess (e.g., Pilis Hills, Balaton Uplands), while they were less frequent on silicate rocks such

as andesite (Visegrád Hills) or rhyolite tuff (e.g., Bükk Hills), but they were also found on the rare marble substrate in the Cserehát (Table 1).

3.  *T. glabrescens* samples were also collected from sites with limestone base rocks (e.g., Buda Hills), dolomite (e.g., Keszthely Hills), or sandy loess (Pilis Hills); however, this species survived in rather extreme conditions, provided by thin bare soil layers developed on basalt, granite, or sandstone rocks (Table 1).

4.  *T. praecox* (TPR) and *T. pulegioides* (TPU) have special ecological preferences: the existence of TPR (specialist, eight populations, 10.81%) populations is connected to soils on calciferous base rocks (dolomite, limestone), while TPU (nine populations, 12.16%) prefers humid areas of mountain and lowland meadows (Tables 1 and 2).

5.  Populations of *T. serpyllum* (TSE), however, were found only in two habitats in our studies, as natural competitor species in plant communities developed on acidic (Somogy Hills) or calciferous sandy soils (Bakony Hills) (Tables 1 and 2).

6.  Where the habitat conditions were favorable, the coexistence of 1–2 thyme species was also observed in the combination of TPA/TGL/TPR (dry conditions, calciferous rocks, and grassland communities) or of TPA/TPU (humid conditions and meadows). In our studies, the most important plant communities involving wild thymes were grasslands on sand, loess, silicate stones, dolomite, and limestones (Table 1).

### 3.2. Essential Oil Levels in Thymus Populations

3.2.1. Essential Oil Production of *Thymus* Species in Different Habitats

In the case of *T. pannonicus,* a high overall mean essential oil content was detected (0.46 mL/100g DW), and the values ranged between the minimum of 0.01 mL/100g (4_14: Szőc, Bakony Hills) and the maximum of 1.90 mL/100 g (8A_27a: Nagykovácsi, Nagy-Szénás, Buda Hills) (Figure 3, Table 3). The latter value represented the highest drug quality with respect to TPA as well as all the *Thymus* samples examined in this experiment. Considering regional mean EO values of TPA samples, almost all the study areas can be recommended for collection, with the exception of the populations surveyed in the Gerecse, Zemplén, and Aggtelek Hills, which showed lower EO values than expected by Ph. Eur. (<0.3 mL/100 g DW) (Figures 3 and 4; Table 3).

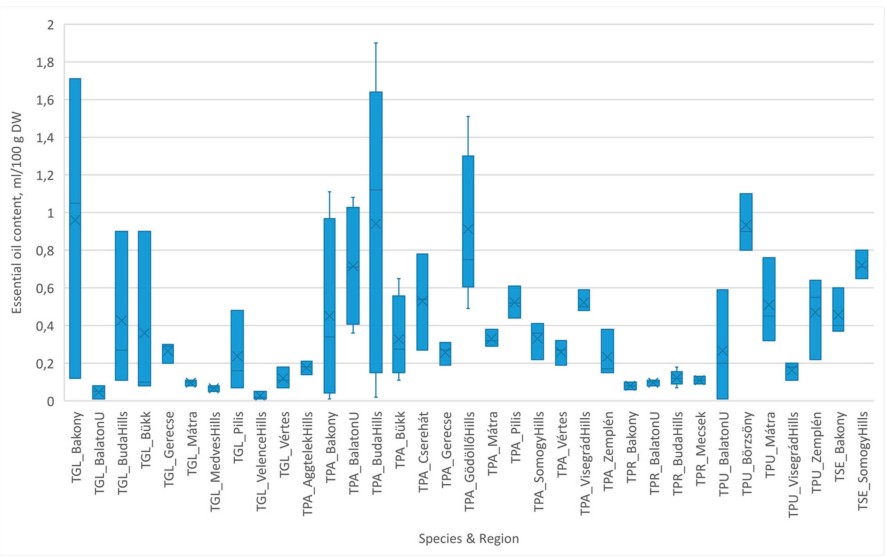

**Figure 3.** Essential oil contents (mL/100 g DW, mean±SD) of samples originating from *Thymus* spp. populations in natural habitats, sorted by different regions surveyed in Hungary. Legends: Abbreviations of species names: TGL: *T. glabrescens*; TPA: *Thymus pannonicus*; TPR: *T. praecox*; TPU: *T. pulegioides*; TSE: *T. serpyllum*.

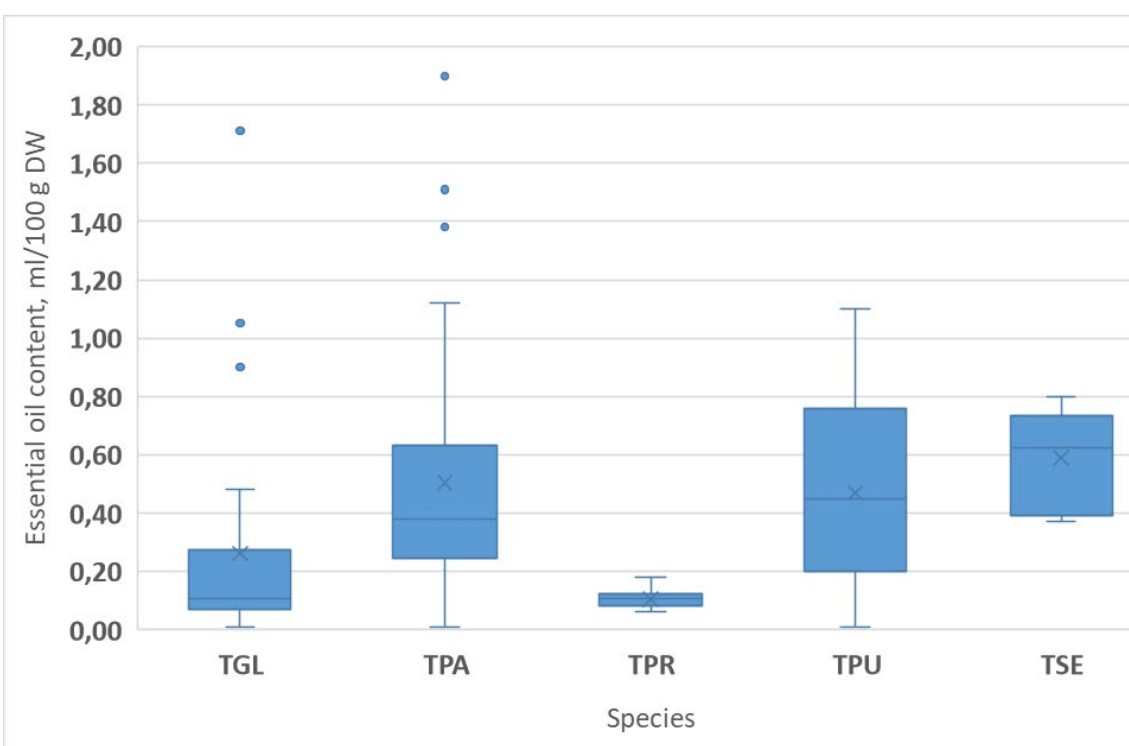

**Figure 4.** Essential oil content (mL/100 g DW, mean±SD) of different *Thymus* species populations surveyed in natural habitats belonging to different regions in Hungary. Legends: Abbreviations of species names: TGL: *T. glabrescens*; TPA: *Thymus pannonicus*; TPR: *T. praecox*; TPU: *T. pulegioides*; TSE: *T. serpyllum*.

**Table 3.** Essential oil content (mL/100 g) and chemotypes determined in native *Thymus* populations in Hungary (2000–2020).

| Site Code, Region_Location ** | Species * Found | Mean EO Content, mL/100 g DW | Relative Percentages (%) of Chief Essential Oil Compounds of Chemotypes | | | | Terpene Class |
|---|---|---|---|---|---|---|---|
| | | | 1st | 2nd | 3rd | 4th | M/MS/S ** |
| 1_1 | TPR | 0.11 | Caryophyllene oxide (28.50) | β-Cubebene (18.00) | | | S |
| 2_2 | TSE | 0.65 | τ-Cadinol (11.80) | Caryophyllene oxide (11.20) | β-Cubebene (8.10) | | S |
| 2_3 | TPA | 0.36 | Thymol (67.50) | | | | M |
| 3_4 | TGL | 0.05 | Thymol (32.88) | β-Caryophyllene (16.50) | Germacrene D (17.62) | | MS |
| 3_5 | TPA | 1.08 | Thymol (51.10) | p-Cymene (13.00) | | | M |
| 3_6 | TPU | 0.59 | Carvacrol (32.20) | Thymol methyl ether (12.10) | γ-Terpinene (9.30) | | M |
| 3_7 | TPA | 0.36 | Thymol (38.25) | p-Cymene (12.07) | β-Bisabolene (11.43) | | MS |
| 3_8a | TPU | 0.01 | β-Caryophyllene (24.50) | Thymol (17.40) | Germacrene D (16.80) | | MS |
| 3_8b | TGL | 0.01 | Thymol (32.90) | β-Caryophyllene (16.50) | Germacrene D (13.50) | | MS |
| 3_9 | TPR | 0.08 | β-Cubebene (27.80) | Caryophyllene oxide (20.29) | β-Caryophyllene (14.70) | | S |
| 3_10 | TPA | 0.87 | Thymol (53.89) | p-Cymene (10.56) | | | M |
| 3_11 | TPA | 0.55 | Thymol (63.70) | p-Cymene (11.50) | | | M |
| 4_12a | TPA | 1.10 | Thymol (40.00) | γ-Terpinene (20.20) | p-Cymene (14.70) | | M |
| 4_12b | TSE | 0.37 | Geranyl isobutyrate (44.00) | 1,8-Cineol (16.90) | | | M |
| 4_13 | TGL | 1.71 | Thymol (34.0) | p-Cymene (22.9) | | | M |
| 4_14 | TPA | 0.01 | Germacrene D (49.00) | β-Farnesene (8.00) | δ-Cadinene (8.00) | | S |
| 4_15a | TPR | 0.06 | Germacrene D (43.90) | β-Caryophyllene (10.60) | | | S |
| 4_15b | TGL | 0.12 | Germacrene D (55.40) | β-Caryophyllene (14.80) | | | S |
| 4_16 | TPA | 0.54 | Thymol (27.70) | Linalyl acetate (18.80) | γ-Terpinene (18.60) | | M |
| 4_17 | TPA | 0.14 | Germacrene D (43.40) | β-Caryophyllene (15.00) | | | S |
| 5_18 | TGL | 0.02 | nd | nd | nd | | nd |
| 6_19 | TGL | 0.18 | Geraniol (49.00) | Germacrene D (13.60) | | | MS |
| 6_20a | TPA | 0.32 | Thymol (30.17) | p-Cymene (26.00) | Thymol methylether (13.38) | γ-Terpinene (9.55) | M |
| 6_20b | TGL | 0.07 | τ-Cadinol (43.20) | Germacrene D (15.55) | cis-γ-Cadinene (10.41) | | S |
| 7_21a | TPA | 0.27 | p-Cymene (53.70 | Geraniol (15.80) | | | M |
| 7_21b | TGL | 0.30 | p-Cymene (45.00) | Geraniol (13.60) | Linalyl acetate (9.9) | | M |
| 8A_22a | TPA | 0.28 | Thymol (38.30) | p-Cymene (17.20) | | | M |
| 8A_22b | TPR | 0.11 | Geraniol (23.20) | Germacrene D (14.70) | β-Caryophyllene (12.20) | | MS |
| 8A_23 | TGL | 0.27 | Germacrene D (43.75) | Thymol (25.03) | | | MS |
| 8A_24a | TPA | 1.10 | Thymol (41.30) | p-Cymene (19.20) | | | M |
| 8A_24b | TPR | 0.01 | Geraniol (18.20) | Germacrene D (16.60) | | | MS |
| 8A_25 | TPA | 0.22 | Thymol (36.50) | p-Cymene (27.30) | | | M |
| 8A_26a | TPA | 1.37 | Geraniol (25.30) | γ-Terpinene (24.70) | | | M |
| 8A_26b | TPR | 0.07 | Germacrene D (31.70) | β-Caryophyllene (21.20) | | | S |
| 8A_27a | TPA | 1.90 | Germacrene D (29.7) | β-Caryophyllene (22.00) | Farnesol (10.40) | | S |
| 8A_27b | TPR | 0.13 | Germacrene D (31.90) | β-Caryophyllene (22.30) | | | S |
| 8A_28 | TGL | 0.90 | Germacrene D (44.73) | β-Caryophyllene (13.88) | Bicyclogermacrene (10.45) | | S |
| 8A_29 | TPR | 0.12 | Germacrene D (39.90) | β-Caryophyllene (21.60) | | | S |
| 8A_30 | TGL | 0.11 | Germacrene D (56.40) | β-Caryophyllene (25.07) | | | S |
| 8A_31 | TPR | 0.18 | Caryophyllene oxide (16.00) | β-Caryophyllene (11.70) | | | S |
| 8B_32 | TGL | 0.16 | Germacrene D (17.80) | Nerolidol (12.99) | β-Cadinene (12.86) | β-Bisabolene (9.19) | S |
| 8B_33 | TPA | 0.52 | Thymol (53.58) | p-Cymene (10.52) | γ-Terpinene (9.63) | | M |
| 8B_34 | TGL | 0.07 | β-Caryophyllene (29.77) | Germacrene D (23.82) | β-Cadinene (11.90) | | S |
| 8B_35 | TGL | 0.48 | Thymol (69.28) | γ-Terpinene (18.28) | | | M |

**Table 3.** *Cont.*

| Site Code, Region_Location ** | Species * Found | Mean EO Content, mL/100 g DW | Relative Percentages (%) of Chief Essential Oil Compounds of Chemotypes | | | | Terpene Class |
|---|---|---|---|---|---|---|---|
| | | | 1st | 2nd | 3rd | 4th | M/MS/S ** |
| 9_36 | TPA | 0.48 | *p*-Cymene (45.13) | Thymol (20.48) | | | M |
| 9_37 | TPU | 0.11 | Germacrene D (21.70) | β-Caryophyllene (13.80) | γ-Muurulene (10.30) | | S |
| 9_38 | TPA | 0.59 | Thymol (52.92) | γ-Terpinene (13.52) | *p*-Cymene (10.72) | | M |
| 9_39 | TPA | 0.50 | Thymol (66.00) | *p*-Cymene (10.27) | | | M |
| 10_40 | TPU | 0.80 | *p*-cymene (18.70) | Spathulenol (16.10) | Geraniol (14.00) | | MS |
| 11_41 | TPA | 0.72 | Thymol (32.00–56.00) | *p*-Cymene (9.80–21.50) | γ-Terpinene (5.90–15.40) | | M |
| 11_42 | TPA | 1.15 | Thymol (43.00–60.00) | *p*-Cymene (6.90–21.10) | γ-Terpinene (7.30–18.50) | | M |
| 11_43 | TPA | 0.83 | Thymol (38.90) | *p*-Cymene (8.90) | γ-Terpinene (11.10) | | M |
| 11_44 | TPA | 1.09 | Thymol (48.00–53.00) | *p*-Cymene (5.90–15.80) | γ-Terpinene (7.10–11.00) | | M |
| 11_45 | TPA | 0.49 | Thymol (32–56) | *p*-Cymene (2.40–6.30) | γ-Terpinene (6.70–7.80) | | M |
| 12_46 | TGL | 0.05 | Thymol (14.40) | Germacrene D (12.10) | Geraniol (10.80) | | MS |
| 13_47 | TGL | 0.08 | Thymol (29.30) | Germacrene D (14.20) | | | MS |
| 13_48a1 | TPU-L | 0.32 | Geranial (22.20) | Linalyl acetate (19.8) | Neral (14.30) | Linalool (14.20) | M |
| 13_48a2 | TPU-T | 0.76 | Thymol (56.20) | γ-Terpinene (10.40) | Thymol methylether (9.90) | | M |
| 13_49 | TPA | 0.32 | Thymol (41.9) | *p*-Cymene (20.2) | Borneol (10.30) | | M |
| 14_50 | TGL | 0.10 | Germacrene D (9.40) | β-Caryophyllene (6.90) | *cis*-Ocymene (6.00) | | MS |
| 14_51 | TPA | 0.27 | β-Cadinene (28.82) | Germacrene D (13.18) | | | S |
| 14_52 | TPA | 0.11 | Linalool (24.44) | *p*-Cymene (14.14) | Thymol (10.78) | Carvacrol (10.31) | M |
| 14_53 | TPA | 0.28 | Linalool (47.12) | *p*-Cymene (*15.18*) | | | M |
| 14_54 | TPA | 0.65 | Carvacrol (40.71) | *p*-Cymene (15.97) | γ-Terpinene (13.67) | | M |
| 15A_55 | TPU | 0.22 | β-Caryophyllene (53.2) | β-Cubebene (19.20) | | | S |
| 15A_56a | TPA | 0.15 | β-Caryophyllene (48.70) | β-Cubebene (19.90) | Thymol (8.00) | | MS |
| 15A_56b | TPU | 0.64 | Linalool (38.1) | Geraniol (23.90) | Linalyl acetate (14.40) | | M |
| 15A_57 | TPA | 0.17 | Caryophyllene oxide (45.10) | β-Cubebene (15.70) | Linalool (13.80) | | MS |
| 15A_58a | TPA | 0.38 | β-Cubebene (24.50) | Linalool (7.59) | Linalyl acetate (7.41) | | MS |
| 15A_58b | TPU | 0.55 | Geraniol (27.50) | Linalyl acetate (20.20) | Thymol methyl ether (13.50) | | M |
| 15B_59 | TPA | 0.14 | Germacrene D (26.35) | Caryophyllene oxide (10.43) | | | S |
| 15B_60 | TPA | 0.21 | Geranyl acetate (24.08) | β-bisabolene (16.27) | Geraniol (12.00) | | MS |
| 15C_61 | TPA | 0.78 | *p*-Cymene (44.90) | Thymol (20.22) | γ-Terpinene (10.12) | | M |
| 15C_62 | TPA | 0.54 | Linalool (26.63) | Thymol (22.25) | *p*-Cymene (14.56) | γ-Terpinene (11.05) | M |
| 15C_63 | TPA | 0.27 | Geraniol (12.67) | Geranyl acetate (11.58) | *p*-Cymene (11.08) | Carvacrol (8.80) | M |

Legends: * Abbreviations of species names: TGL: *T. glabrescens*; TPA: *Thymus pannonicus*; TPR: *T. praecox*; TPU: *T. pulegioides*; TSE: *T. serpyllum* ** Abbreviations of chemo variant classes: M: monoterpene chief compounds only; MS: mono-and sesquiterpene chief compounds; S: sesquiterpene chief compounds only. nd: not detected.

The average essential oil content (0.26 mL/100 g) of *T. glabrescens* samples surveyed was lower than the minimum value of the pharmacopoeial standard; however, in some cases (Bakony, Gerecse, and Buda Hills), the collected drug may fulfil the requirements. The highest essential oil content (1.71 mL/100 g) of *T. glabrescens* was measured in the sample originating from Csesznek, Castle Hill (4_13, Bakony Hills), while the poorest quality (0.01 mL/100 g) was obtained among the extreme habitat conditions of Stone Hill, Szentbékkálla (3_8B: Balaton Uplands) (Figure 3, Table 3).

Concerning the essential oil content of *T. praecox*, our results always indicated substandard drug quality, where the values ranged between 0.06 mL/100 g (4_15a: Várpalota, Bakony Hills) and 0.18 mL/100 g (8A_31: Diósd, Buda Hills), with an average of 0.1 mL/100 g. None of these populations are suggested for collecting flowering shoots of TPR to obtain dried drugs (Figure 3, Table 3).

The essential oil content of *T. pulegioides* samples varied between very low (0.01 mL/100 g: 3_8A, Szentbékkálla, Balaton Uplands) and high (0.8 mL/100 g: 10_40, Szent Mihály Hill, Börzsöny Hills:) values (mean: 0.47 mL/100 g). The accumulation levels were usually higher than 0.3 mL/100 g, except for the sample originating from the Visegrád Hills (site no. 9_37) (Figure 3, Table 3).

*T. serpyllum* was the only species where both samples (2_2: Nagybajom, Somogy Hills, and 4_12B: Fenyőfő, Bakony Hills) fulfilled the requirement of the Parmacopoeial standard, and the average was also rather high (0.59 mL/100g) (Figure 3, Table 3).

### 3.2.2. The Role of the Genetic Factor in the Essential Oil Accumulating Ability

In our studies, the effect of the genotype was significant for the essential oil content detected in populations belonging to different wild thyme species, according to the ANOVA ($p < 0.001$). This phenomenon seemed to be highly evident when 1–3 species occur among equal circumstances in the same habitat, but rather different EO levels can be detected in the drug (e.g., see Table 3: 8A_24a: TPA:1.10 mL/100 g; 8A_24b: TPR: 0.01 mL/100 g). In our experiment, natural populations belonging to TSE, TPA, and TPU species provided

adequate amounts of essential oil. TSE: 0.59 ± 0.17 mL/100g TPA: 0.50 ± 0.39 mL/100 g, TPU: 0.47 ± 0.32 mL/100 g, while TGL (0.26 ± 0.28 mL/100 g) and TPR (0.11 ± 0.03 mL/100 g) accumulated rather low levels (Figures 3 and 4). *T. praecox* showed the poorest drug quality (group 'a'); the difference was statistically proven with respect to *T. pannonicus* and *T. serpyllum*, both of which belonged to group 'b'. *T. glabrescens* and *T. pulegioides* represented a transitional group ('ab') that is quite variable in essential oil levels (Figure 4).

When comparing the performance of the 5 species studied in essential oil accumulating, we can conclude that most of the samples (35) can be categorized into the range of 0.1–0.5 mL/100 g or into the group of 0.5–1.0 mL/100 g (18) (Figure 5). Only a few populations showed excellent levels of essential oils (1.0–1.5 mL/100 g: 5; 1.5 < mL/100 g: 3), while the number of cases with very low EO quantities was higher (13). As far as the species is concerned, TPA, TPU, and TSE possessed appropriate essential oil contents with higher frequency, while TGL was very diverse, and TPR can be completely excluded from collection based on low accumulation levels (Figure 5).

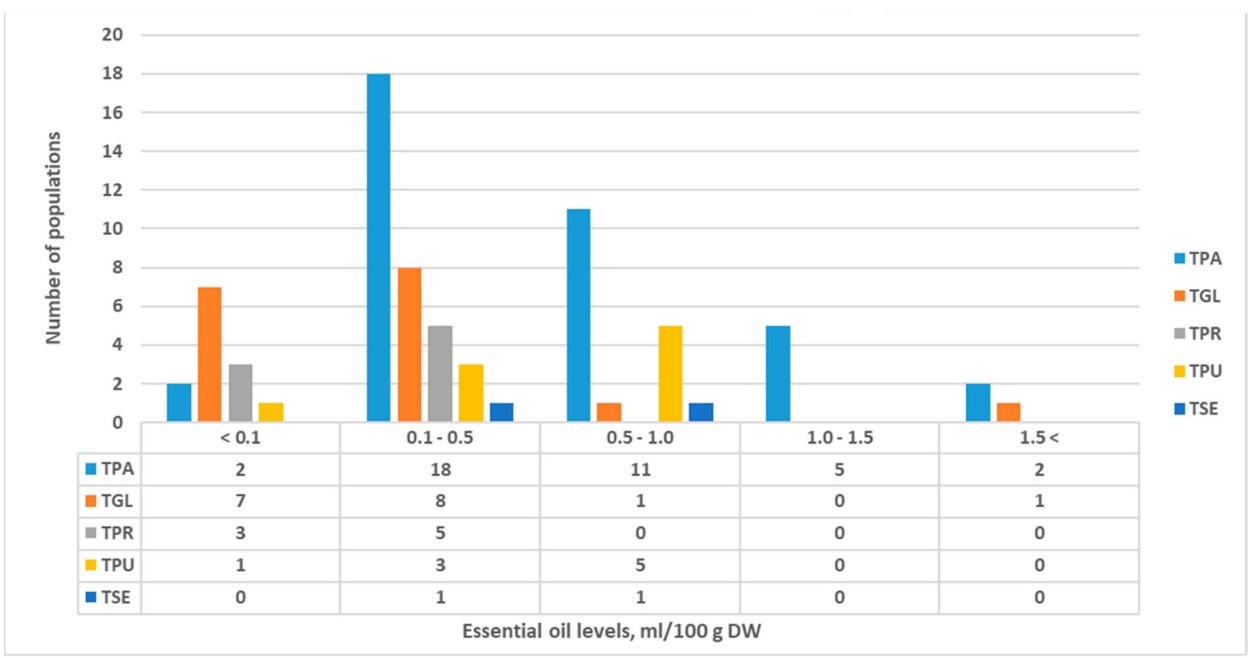

**Figure 5.** Frequency of essential oil accumulation levels (mL/100 g DW) in *Thymus* spp. Populations representing different species in Hungary. Legends: Abbreviations of species names: TGL: *T. glabrescens*; TPA: *Thymus pannonicus*; TPR: *T. praecox*; TPU: *T. pulegioides*; TSE: *T. serpyllum*.

### 3.2.3. The Role of the Environmental Factors on the Essential Oil Accumulating Ability

Environmental conditions highly affected the essential oil levels measured, as shown by the average values originating from different regions involved (Figure 6). The statistical analysis verified that the differences among the regions were significant ($p = 0.008$). In general, Somogy Hills (TPA, TSE), Buda Hills (TPA), Börzsöny Hills (TPU), Gödöllő Hills (TPA), and Cserehát Hills (TPA) have proven to be the most valuable areas (EO > 0.50 mL/100 g DW) to collect and produce wild thyme drugs. If considering the EO requirement of (>0.3 mL/100 g DW) of the Ph. Eur. standard, seven regions (Mecsek, Velence, Vértes, Gerecse, Medves, Bükk, and Aggtelek Hills) are to be discarded according to the substandard (<0.30 mL/100 g DW) mean values (Figure 6). However, the data obtained from smaller locations and populations belonging to certain thyme species may represent proper drug quality.

Very low essential oil levels have been detected in thyme populations situated on exposed rock surfaces (e.g., gray limestone in Mecsek Hills; granite rocks in Velence Hills; Pannonian sandstone rocks at Szentbékkálla or andesite cliffs at Dömös, etc.), covered sometimes only by thin bare soil layers, which are considered an extreme condition. In these places, secondary metabolite production can be highly restricted.

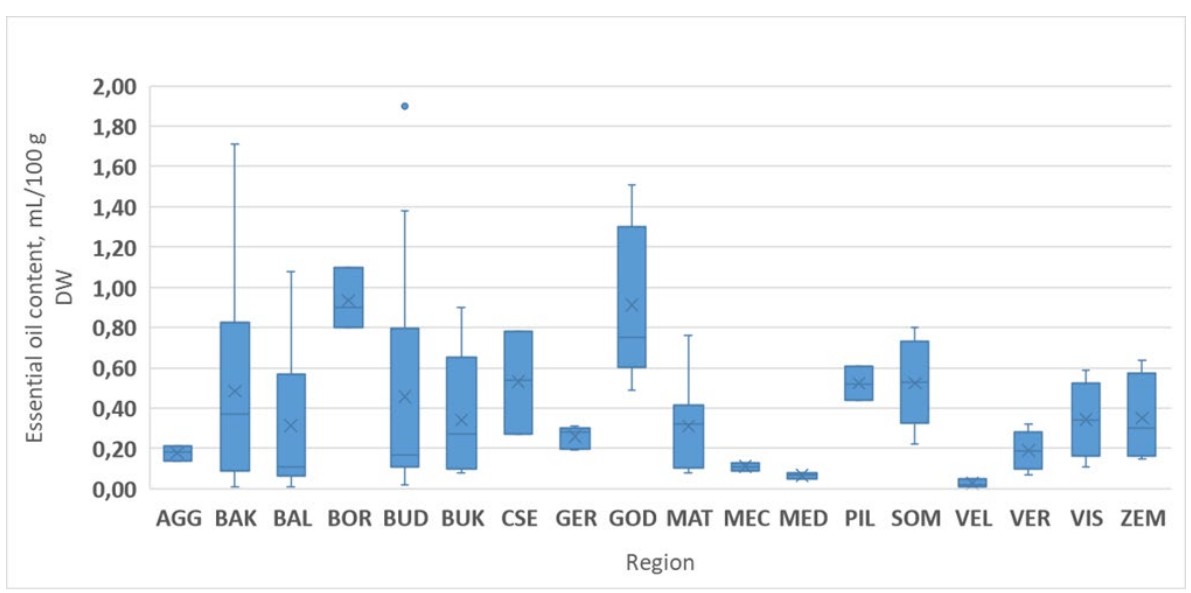

**Figure 6.** Essential oil accumulation levels (mL/100 g DW) of *Thymus* spp. populations in different regions of Hungary. Legends: Abbreviations of region names: AGG: Aggtelek Hills; BAK: Bakony Hills; BAL: Balaton Uplands; BOR: Börzsöny Hills; BUD: Buda Hills; BUK: Bükk Hills; CSE: Cserehát Hills; GER: Gerecse Hills; GOD: Gödöllő Hills; MAT: Mátra Hills; MEC: Mecsek Hills; MED: Medves Hills; PIL: Pilis Hills; SOM: Somogy Hills; VEL: Velence Hills; VER: Vértes Hills; VIS: Visegrád Hills; ZEM: Zemplén Hills.

### 3.3. Essential Oil Chemotypes in Native Thymus Populations

#### 3.3.1. Thymus pannonicus

A high level of essential oil diversity in *T. pannonicus* was proven on the basis of samples collected from wild populations. According to the chief compounds of their essential oils, the Hungarian thyme populations investigated could be classified into 18 well-defined chemotypes (Figure 7, Table 3). Chemotypes were basically monoterpene-dominated (28 populations, 73.70%), while the proportions of sesquiterpene types (15.80%) and combined ones (10.53%) were rather low.

Among the monoterpenes, thymol (8.00–66.00%) played an important role in influencing the essential oil quality as the chief compound of 25 populations of different origins, especially in the Transdanubian Mountain range (W-Hungary) (Figure 7, Table 3). Other significant monoterpenes were the related *p*-cymene (8.90–43.15%; 26 sites) and γ-terpinene (9.63–24.70%; 12 sites); however, linalool (7.59–47.12%; 5 sites), geraniol (12.00–25.30%; 4 sites), carvacrol (10.31–40.70%; 3 sites), linalyl acetate (7.41–18.80%; 2 sites), and geranyl acetate (15.58–24.08%; 2 sites) were also found in several EOs, respectively. Of the sesquiterpenes, the highest frequency was detected at germacrene D (13.18–49.00%; five populations), *β*-caryophyllene (15.00–48.70%; three populations), and *β*-cubebene (13.80 23.40%; three populations). However, other compounds (e.g., caryophyllene oxide, *β*-bisabolene, *β*-farnesene, *β*-cadinene, and farnesol) also reached considerable levels in certain essential oil samples (Figure 7, Table 3).

Concerning TPA populations, 11 new sites were studied, and monoterpene (M) chief EO compounds were found. Among them, eight provided thymol-dominated essential oils (3–5; 3_10, 8B_33; 9_36; 9_38; 9_39; 14_53; 15C_61), while the other three (see below as four, five, and eight) could be considered new chemotypes with carvacrol (Bükk Hills, Mónosbél), geraniol+carvacrol (Cserehát Hills, Sajógalgóc), or linalool+thymol+carvacrol (Bükk Hills, Noszvaj) chief EO compounds (Figure 7, Table 3).

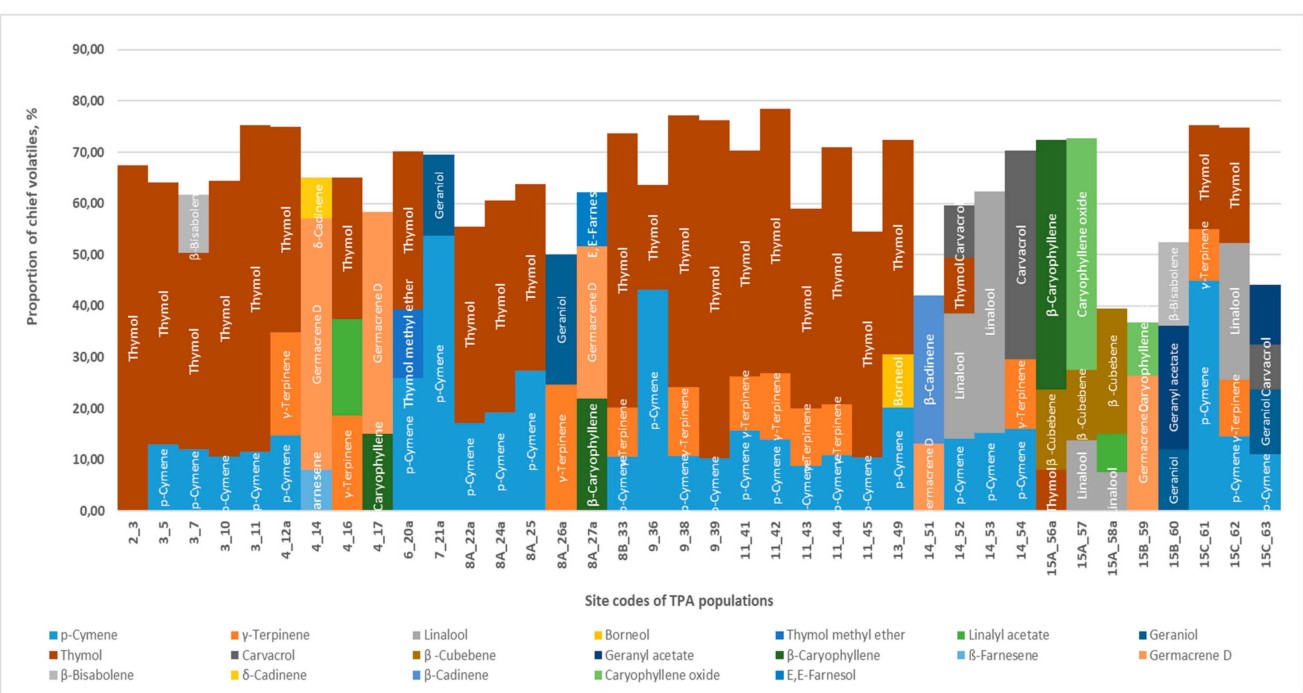

**Figure 7.** The chemotype pattern of *Thymus pannonicus* essential oils originating from native populations exists in different habitats in Hungary. (Legends: see site codes in Table 1).

Combined chemotypes involved thymol, geraniol, or linalool as chief monoterpene molecules, while the most frequent sesquiterpene partners were *β*-bisabolene and *β*-cubebene, respectively. In the latter group, chemotypes no. 9 and 11 showed new compositions, where *β*-bisabolene represented the S fraction, while thymol (Balaton Uplands, Tapolca) or geraniol (Aggtelek Hills, Jósvafő) were the compounds with M structure (Figure 7, Table 3).

Germacrene D was the main non-oxygenated sesquiterpene in S-dominated chemotypes, where four new EO compositions (No. 15, 16, 17, and 18) were recorded (Figure 7, Table 3).

The chemotype patterns detected in TPA essential oil samples with the respective data are summarized as follows (new data are of bold type):

Monoterpene chemotype (M):

1. Thymol chemotype (+ *γ*-terpinene + *p*-cymene) (18 sites)—limestone, loess, sand, andesite, and dolomite.
2. Thymol + *γ*-terpinene+linalyl acetate (Bakony Hills, Öskü)—dolomite.
3. Thymol + *p*-cymene +isoborneol (Mátra Hills, Sirok)—rhyolite tuff.
4. Carvacrol + *γ*-terpinene + *p*-cymene (Bükk Hills, Mónosbél)—mudstone.
5. Geraniol + geranyl acetate + *p*-cymene + carvacrol (Cserehát Hills, Sajógalgóc)—mudstone.
6. Geraniol + *p*-cymene (Gerecse Hills, Tardosbánya)—limestone.
7. Geraniol + *γ*-terpinene (Buda Hills, Homok Hill)—dolomite.
8. Linalool + *p*-cymene + thymol + carvacrol (Bükk Hills, Noszvaj)—mudstone.

Combined chemotype of mono-and sesuiterpenes (MS):

9. Thymol +*p*-cymene + *β*-bisabolene (Balaton Uplands, Tapolca)—limestone.
10. Thymol + *β*-caryophyllene + *β*-cubebene (Zemplén Hills, Regéc hayfield)—rhyolite.
11. Geraniol + geranyl acetate + *β*-bisabolene (Aggtelek Hills, Jósvafő)—limestone.
12. Linalool + linalyl acetate + *β*-cubebene (Zemplén Hills, Vágáshuta)—andesite.
13. Linalool + caryophyllene oxide + *β*-cubebene (Zemplén Hills, Bózsva)—rhyolite.

Sesuiterpenes chemotype (S):

14. Germacrene D + *β*-caryophyllene (Bakony Hills, Litér)—dolomite.
15. Germacrene D + *β*-caryophyllene + farnesol (Buda Hills, Nagyszénás)—dolomite.
16. Germacrene D + caryophyllene oxide (Aggtelek Hills, Aggtelek)—limestone.

17.    Germacrene D + *β*-farnesene + *δ*-cadinene (Bakony Hills, Szőc)—limestone
18.    Germacrene D + *β*-cadinene (Bükk Hills, Cserépváralja)—rhyolite tuff.

### 3.3.2. *Thymus glabrescens*

In the case of *T. glabrescens,* EO samples of different origins contained a wide spectra of sesquiterpene chief compounds: germacrene D, *β*-caryophyllene, *τ*-cadinol, or *β*-cadinene, respectively (Figure 5, Table 3). Among monoterpenes, thymol (7) or geraniol (3) were determined with the highest frequency, while the most abundant sesquiterpenes in the chemotype patterns were germacrene-D (13) and *β*-caryophyllene (7). Both combined (MS) and sesquiterpene (S) chemotypes occurred in 6-6 populations, while monoterpene (M) ones were detected only in four cases, respectively (Figure 8, Table 3). Thymol was detected in the range of 14.40–69.28 %, where the highest level was found at Pilis Hill at Pilisszántó (8B_35) in a monoterpene chemotype. Geraniol was present in the EOs of monoterpene (13.60%) or in combined (49.00%; 10.80%) chemotypes, while the percentages of the generally occurring in germacrene-D varied from 12.10 to 56.40%, respectively.

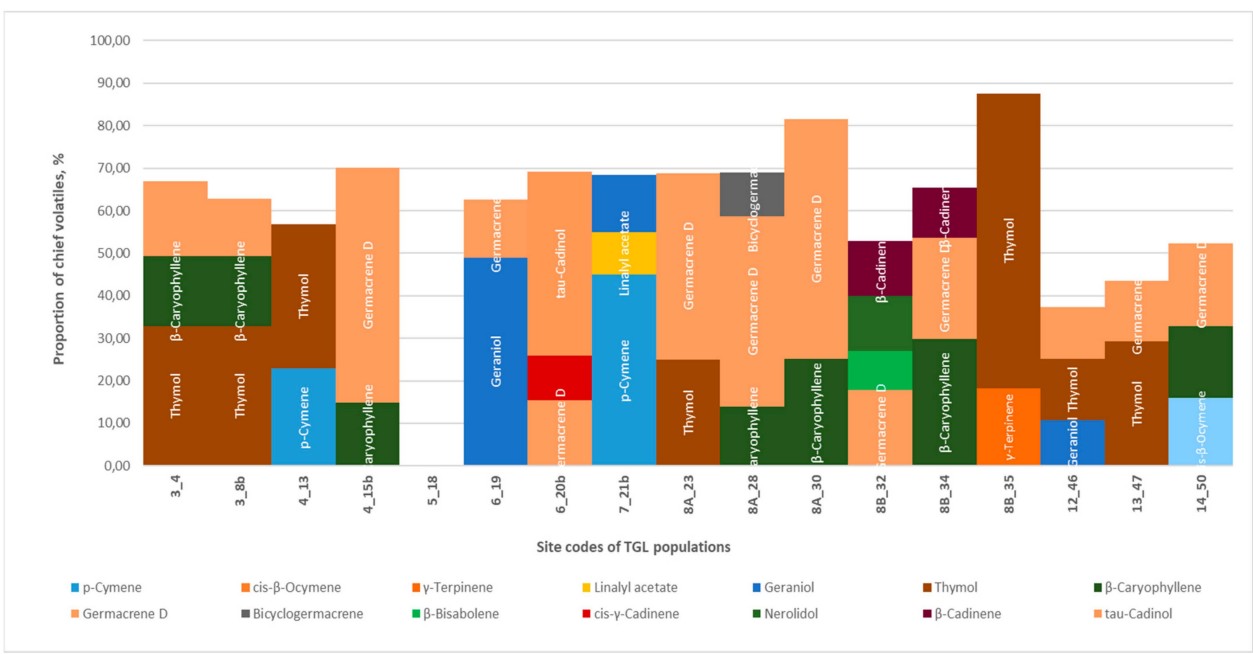

**Figure 8.** Chemotype pattern of *Thymus glabrescens* essential oils originating from native populations exist in different habitats of Hungary. (Legends: see site codes in Table 1).

The following 12 chemotypes are described below (Figure 8, Table 3) (data of new surveys are of bold type):
Monoterpene chemotype (M):

1.    Thymol + *γ*-terpinene (Pilis Hills, Pilisszántó)—dolomite.
2.    Thymol + *p*-cymene (Bakony Hills, Csesznek)—limestone.
3.    Geraniol + *p*-cymene + linalyl acetate (Gerecse: Tardosbánya)—limestone.

Combined chemotype of mono-and sesuiterpenes (MS):

4.    Thymol + geraniol + germacrene D (Medves Hills, Salgó Hill)—basalt.
5.    Thymol + germacrene D (Buda Hills: Kálvária Hill; Mátra Hills: Pásztó, Köves Cliff).
6.    Thymol + germacrene D + *β*-caryophyllene (Balaton Uplands, Várvölgy; Szentbékkálla)—sand; sandstone.
7.    Cis-*β*-Ocymene + germacrene D + *β*-caryophyllene (Bükk Hills, Szarvaskő)—mudstone.

Sesquiterpenes chemotype (S):

8. Germacrene D + β-caryophyllene (Bakony Hills, Várpalota; Buda Hills, Érd)—dolomite, limestone.
9. Germacrene D + β-caryophyllene +bicyclogermacrene (Buda Hills, Nagykovácsi Dog Hill)—limestone.
10. Germacrene D+ β-caryophyllene + β-cadinene (Pilis Hills, Pilisszentiván, Fehér Hill)—dolomite.
11. Germacrene D + nerolidol + β -cadinene (Pilis Hills, Dorog, Strázsa Hill)—Ca-sand.
12. Germacrene D + τ-cadinol (Vértes Hills, Csákberény)—dolomite.

### 3.3.3. *Thymus praecox*

In the essential oils of *T. praecox*, five compounds were significant, determining the chemotype patterns. The monoterpene geraniol (23.20%) and two sesquiterpene hydrocarbons (germacrene D at 14.70–43.90% and β-caryophyllene at 10.60–22.30%) were important constituents in the composition, along with caryophyllene oxide (16.00–28.50%) in three samples and β-cubebene (18.00–27.80%) in two samples (Figure 6, Table 3). Among the eight EO samples of different origins, only one could be considered a combined (MS) chemotype of mono- and sesquiterpene chief compounds (8A_22B: Budaörs, Odvas Hill, Buda Hills). All the others belong to the sesquiterpene class (S), with the highest frequency of germacrene D + β-caryophyllene chemotype (Figure 9, Table 3).

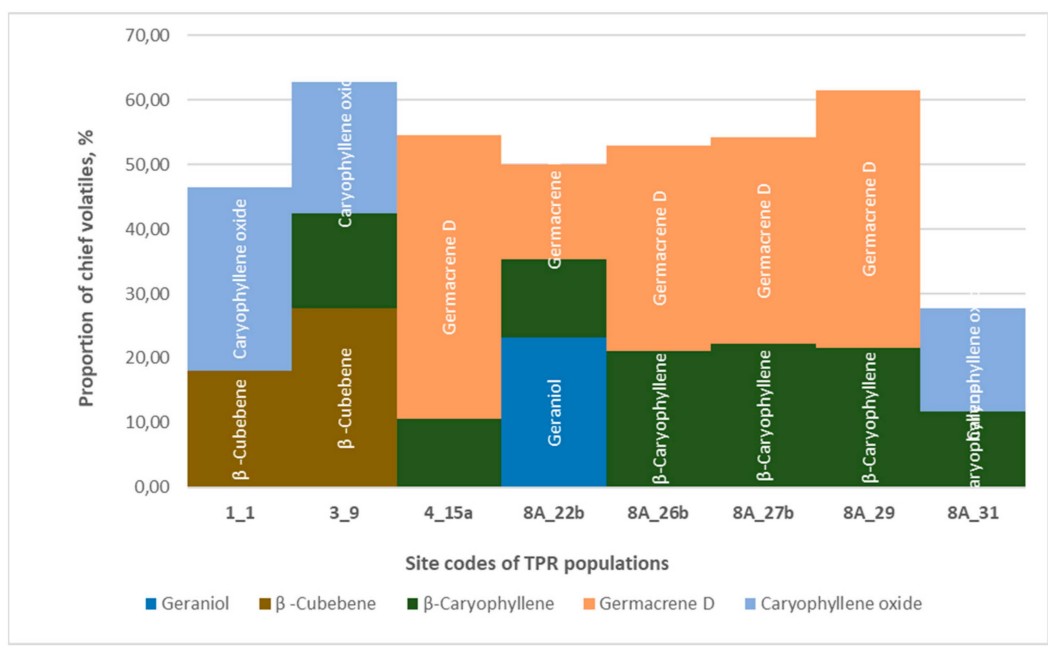

**Figure 9.** Chemotype pattern of *Thymus praecox* essential oils originating from native populations exist in different habitats of Hungary. (Legends: see site codes in Table 1).

Five essential oil chemotypes of TPR are described as follows:
Sesquiterpene (S) chemotypes:

1. Germacrene D + β-caryophyllene (Buda Hills: Sas Hill, Nagyszénás Hill, Homok Hill; Bakony Hills: Várpalota): dolomite.
2. β-caryophyllene + caryophyllene oxide (Buda Hills: Tétény Hill, Diósd): Sarmathian limestone.
3. β-cubebene + caryophyllene oxide (Mecsek Hills: Pécs, Kis-Tubes Hill)—limestone.
4. β-cubebene + caryophyllene oxide + β-caryophyllene Balaton Uplands, Tamás Hill, Balatonfüred): dolomite.

Combined chemotype of mono-and sesuiterpenes (MS):

5. Geraniol + germacrene D + *β*-caryophyllene (Buda Hills, Odvas Hill, Budaörs): dolomite.

### 3.3.4. *Thymus pulegioides*

High levels of essential oil diversity of *T. pulegioides* were proven on the basis of samples collected from wild populations (Figure 10). Fifteen distinct chief compounds were identified in the essential oils of *T. pulegioides* samples of different origins (Table 3). Among monoterpenes, thymol played an important role only in two habitats (3_8a; 13_48a2), while others were also significant in some populations (*p*-cymene: 10_40; *γ*-terpinene: 3_8a, 13_48a2; linalool: 13_48a1, 15a_56b; thymol methyl ether: 3_8a, 13_48a2; neral: 13_48a1; geraniol: 10_40, 15a_56b, 15a_58b; linalyl acetate: 13_48a1, 15a_56b, 15a_58b; geranial: 13_48a1 and carvacrol: 3_8a). Of the sesquiterpenes, the highest percentages were detected at *β*-caryophyllene (3_6, 9_37) and germacrene D (3_6 and 9_37). However, other compounds (e.g., *γ*-muurulene, spathulenol) also reached considerable levels in certain essential oil samples (Figure 10).

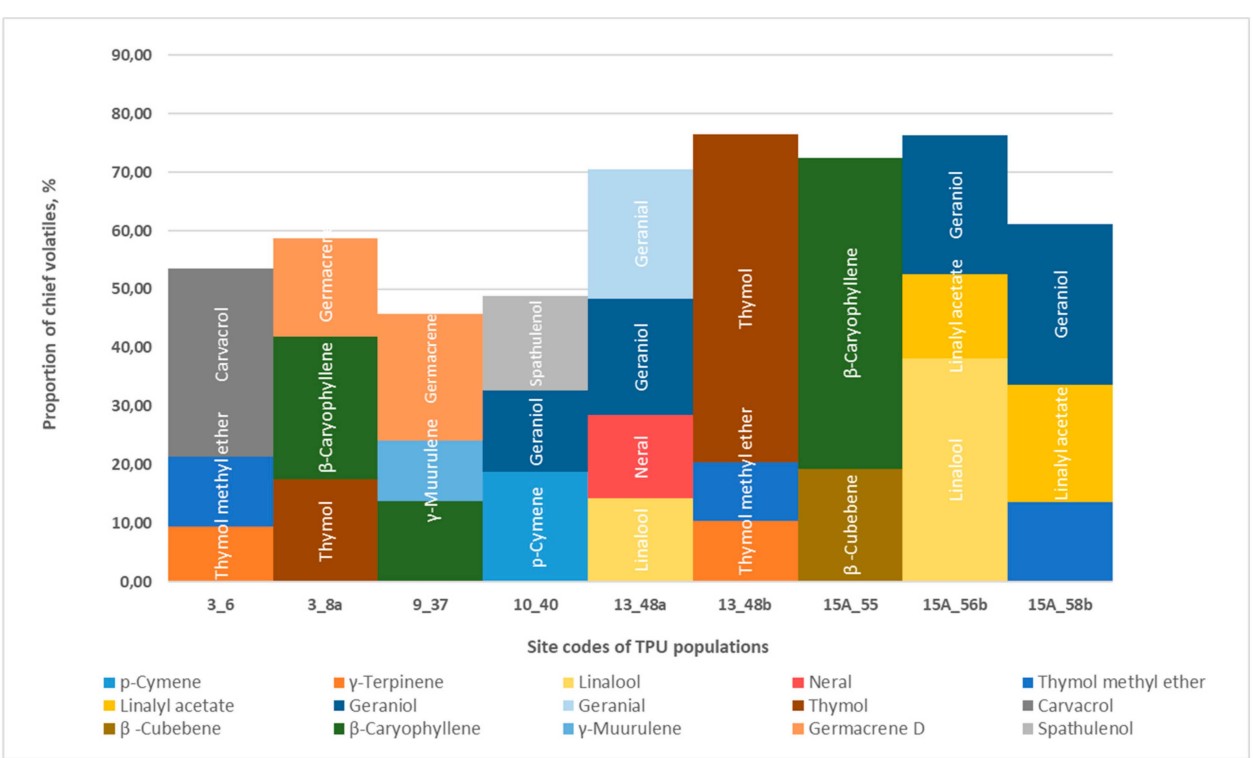

**Figure 10.** Chemotype pattern of *Thymus pulegioides* essential oils originating from native populations exist in different habitats of Hungary. (Legends: see site codes in Table 1).

According to the chief compounds, their essential oil samples could be classified into well-defined monoterpene, sesquiterpene, and mixed chemotypes, which are closely related to the different habitats surveyed. Apart from those known from the previous literature (thymol, linalool/geraniol/linalyl acetate, and geraniol/linalyl acetate), the following five new chemotypes are described below (Figure 10):

Monoterpene chemotype (M) of phenolic character:

1. Carvacrol + thymol metylether + *γ*-terpinene (Balaton Uplands, Zalaszántó)—silt.

Monoterpene chemotype (M) of lemon odour:

2. Geranial + linalyl acetate + neral + linalool (Mátra Hills, Mátrakeresztes)—andesite.

   Combined chemotype of mono-and sesuiterpenes (MS):

3. *p-cymene + spathulenol +geraniol (Börzsöny Hills, Szent Mihály Hill, Zebegény)—andesite.*

4.    *β*-caryophyllene + thymol + germacrene D (Balaton Uplands, Szentbékkálla)—sandstone.

Sesuiterpenes chemotype (S):

5.    Germacrene D + *β*-caryophyllene + *γ*-muurulene (Visegrád Hills, Vadálló Cliffs, Dömös)—andesite.

### 3.3.5. *Thymus serpyllum*

Both of the *T. serpyllum* populations examined represent new chemotypes with special monoterpene (M) and sesquiterpene (S) composition as follows (Figure 11, Table 3):

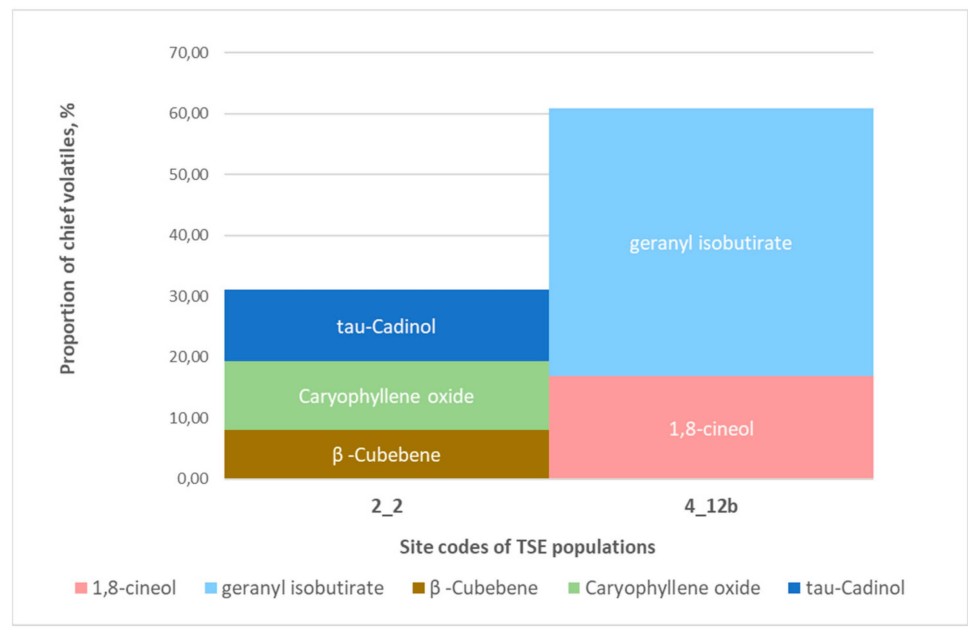

**Figure 11.** Chemotype pattern of *Thymus serpyllum* essential oils originating from native populations exist in different habitats of Hungary. (Legends: see site codes in Table 1).

Monoterpene chemotype (M):

1.    Geraniol + geranyl isobutyrate (Bakony Hills, Fenyőfő)—basic sand.

Sesuiterpenes chemotype (S):

2.    *τ*-cadinol + caryophyllene oxide + *β*-cubebene (Somogy Hills, Nagybajom)—acidic sand.

Frequency of Chief Volatiles Detected in *Thymus* Chemotypes

Concerning the therapeutic aspects of the drugs collected, not only should the essential oil contents be considered, but also the appropriate proportion of effective compounds detected by GC/MS as relative percentages. In the case of the *Thymus* species, utilization of the drugs and industrial raw materials is generally based on monoterpene phenol compounds (thymol, carvacrol) and derivatives (thymol methyl ether, carvacrol methyl ether), as well as biosynthetic intermediates (*p*-cymene, *γ*-terpinene) that are detectable in typical thyme essential oils, where high thymol percentages are expected.

In the EO samples of native *Thymus* populations studied, altogether 30 terpene molecules were identified as the main compounds of essential oil constituting different chemotypes, listed in Table 4 in the order of elution (1–30) during GC analysis. Supplementary data concerning retention time (RT) and linear retention indices (LRI) are also included. Half of these terpenoid molecules (15) were found to belong to the monoterpene group (M), where a wider variability of oxygenated monoterpenes (MO: 12), including thymol, linalool, or geraniol, were detected more than in the non-oxygenated monoterpene class (MH: 3). On the contrary, most of the chemotype-determining sesquiterpenes (S) can be

grouped with the non-oxygenated sesquiterpenes (SH: 11), while fewer oxygenated (SO: 4) ones were detected in wild thyme oil. The former included the very abundant germacrene D found in 25 EOs and β-caryophyllene, which was present in 21 EO samples (Table 4).

**Table 4.** Frequency of occurrence, analytical data and classification of chief essential oil compounds of chemotypes found in native *Thymus* populations in Hungary.

| No. | Compound | RT | LRI | Terpene Class ** | Frequency of Chief Compounds by Species * (Number of Occurrence) | | | | | Overall Freqency of Compounds in EO Samples | | |
|---|---|---|---|---|---|---|---|---|---|---|---|---|
| | | | | | TPA | TGL | TPR | TPU | TSE | Total No. of Occur. | Total Share, % | Mean % |
| 1 | *p*-Cymene | 8.09 | 1026 | MH | 26 | 2 | | 1 | | 29 | 42.00 | 19.39 |
| 2 | 1,8-Cineol | 8.38 | 1034 | MO | | | | | 1 | 1 | 1.35 | 16.90 |
| 3 | cis-β-Ocymene | 8.50 | 1036 | MH | | | 1 | | | 1 | 1.35 | 16.00 |
| 4 | γ-Terpinene | 9.20 | 1056 | MH | 12 | 1 | | 2 | | 15 | 20.27 | 13.61 |
| 5 | Linalool | 10.76 | 1097 | MO | 5 | | | 2 | | 7 | 9.46 | 24.55 |
| 6 | Isoborneol | 13.43 | 1162 | MO | 1 | | | | | 1 | 1.35 | 10.30 |
| 7 | Thymol methyl ether | 16.20 | 1228 | MO | 1 | | | 3 | | 4 | 5.41 | 12.23 |
| 8 | Neral | 16.58 | 1249 | MO | | | | 1 | | 1 | 1.35 | 14.30 |
| 9 | Linalyl acetate | 17.11 | 1250 | MO | 2 | 1 | | 2 | | 5 | 6.76 | 14.14 |
| 10 | Geraniol | 17.20 | 1252 | MO | 4 | 3 | 1 | 4 | | 12 | 16.22 | 20.44 |
| 11 | Geranial | 17.86 | 1268 | MO | | | | | | 1 | 1.35 | 22.20 |
| 12 | Thymol | 18.81 | 1290 | MO | 25 | 7 | | 1 | | 34 | 45.95 | 38.98 |
| 13 | Carvacrol | 19.20 | 1300 | MO | 3 | | | 1 | | 4 | 5.41 | 23.00 |
| 14 | Geranyl acetate | 22.43 | 1388 | MO | 2 | | | | | 2 | 2.7 | 17.83 |
| 15 | β-Cubebene | 22.47 | 1389 | SH | 3 | | 2 | 1 | 1 | 7 | 9.46 | 18.16 |
| 16 | β-Caryophyllene | 23.68 | 1420 | SH | 3 | 7 | 7 | 3 | 1 | 21 | 28.38 | 21.25 |
| 17 | β-Farnesene | 25.27 | 1459 | SH | 1 | | | | | 1 | 1.35 | 8.00 |
| 18 | γ-Muurulene | 25.99 | 1477 | SH | | | | 1 | | 1 | 1,35 | 10.30 |
| 19 | Germacrene D | 26.18 | 1482 | SH | 5 | 13 | 5 | 2 | | 25 | 33.79 | 27.95 |
| 20 | Bicyclogermacrene | 26.81 | 1497 | SH | | 1 | | | | 1 | 1.35 | 10.45 |
| 21 | β-Bisabolene | 27.23 | 1508 | SH | 2 | 1 | | | | 3 | 4.05 | 12.30 |
| 22 | cis-γ-Cadinene | 27.49 | 1515 | SH | | 1 | | | | 1 | 1.35 | 10.41 |
| 23 | δ-Cadinene | 27.80 | 1524 | SH | 1 | | | | | 1 | 1.35 | 8.00 |
| 24 | Geranyl isobutyrate | 29.33 | 1566 | MO | | | | | 1 | 1 | 1.35 | 44.00 |
| 25 | Nerolidol | 29.51 | 1570 | SO | | 1 | | | | 1 | 1.35 | 12.99 |
| 26 | β-Cadinene | 29.87 | 1580 | SH | 1 | 2 | | | | 3 | 4.05 | 17.86 |
| 27 | Spathulenol | 29.98 | 1584 | SO | | | | 1 | | 1 | 1.35 | 16.10 |
| 28 | Caryophyllene oxide | 30.20 | 1590 | SO | 2 | | 3 | | 1 | 6 | 8.11 | 21.92 |
| 29 | τ-Cadinol | 32.26 | 1644 | SH | | 1 | | | 1 | 2 | 2.70 | 43.20 |
| 30 | *E,E*-Farnesol | 35.33 | 1728 | SO | 1 | | | | | 1 | 1.35 | 10.40 |
| | Non-oxygenated monoterpenes (MH) | | | 3 | 38 | 4 | 0 | 3 | 0 | 45 | | |
| | Oxygenated monoterpenes (MO) | | | 12 | 41 | 11 | 1 | 14 | 2 | 73 | | |
| | Total monoterpenes (M) | | | 15 | 79 | 15 | 1 | 17 | 2 | 118 | | |
| | Non-oxygenated sesquiterpenes (SH) | | | 11 | 16 | 26 | 14 | 7 | 3 | 66 | | |
| | Oxygenated sesquiterpenes (SO) | | | 4 | 2 | 1 | 3 | 1 | 1 | 9 | | |
| | Total sesquiterpenes (S) | | | 15 | 18 | 27 | 17 | 8 | 4 | 75 | | |

Legends: * Abbreviations of species names: TGL: *T. glabrescens*; TPA: *Thymus pannonicus*; TPR: *T. praecox*; TPU: *T. pulegioides*; TSE: *T. serpyllum* ** Abbreviations of terpene classes: M: monoterpene; MH: non-oxygenated monoterpene; MO: oxygenated monoterpene; S: sesquiterpene; SH: non-oxygenated sesquiterpene; SO: oxygenated sesquiterpene RT: retention time; LRI: linear retention index.

In our studies, thymol was the most frequent chemotype-determining monoterpene due to the wide distribution and occurrence of *T. pannonicus* populations in Hungary, where this compound dominated the essential oils. Concerning all of the EO samples, thymol (34) and the relative compounds, p-cymene (29) and γ-terpinene (15), could be detected with the highest frequency. Geraniol was also important among oxygenated monoterpenes, represented by three species (TPA, TPU, and TGL). β-caryophyllene was the only terpenoid compound that was detected as a chemotype-determining molecule in all of the five species involved in our studies (Table 4).

The widest spectrum of terpenoids, as possible molecules of the essential oil chemotypes, was found in *T. pannonicus* (19 compounds), followed by *T. pulegioides* (14 compounds), while the lowest variability of chemotypes was shown in *T. praecox* (5 compounds) (Table 4).

If considering the total number of chemotypes detected in all of the samples collected in the populations of *Thymus* species, the number of cases (28) at monoterpene-dominated chemotypes of *T. pannonicus* was proven to be outstanding (Figure 10). We can also add that this high value is principally due to the highly abundant thymol chemotype at TPA. The occurrence of monoterpene chemotypes was much lower in other species (TPU: 4; TGL: 4), while completely absent in *T. praecox*. The dominance of sesquiterpene (S) or combined (MS) chemotypes was characteristic in the samples of *T. praecox* and *T. glabrescens* (Figure 12).

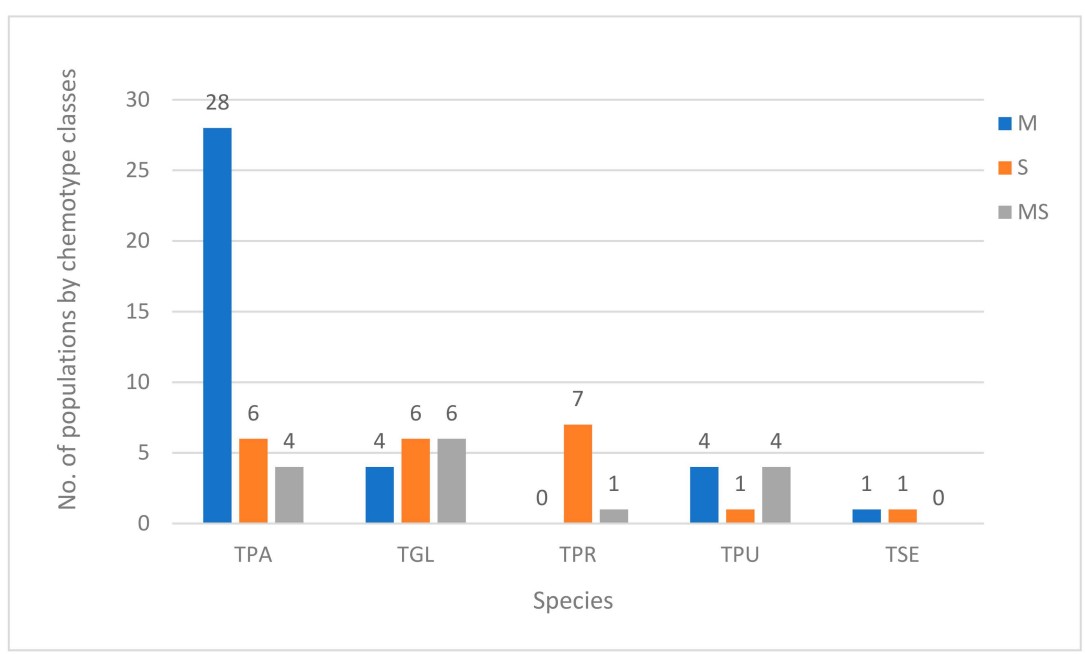

**Figure 12.** Frequency of EO chemotype classes representing different terpene structures in the populations of *Thymus* species in Hungary. Legends: Abbreviations of EO chemotypes by structure of chief compounds: M: monoterpene chief compounds only; MS: mono- and sesquiterpene chief compounds; S: sesquiterpene chief compounds only; Abbreviations of species names: TGL: *T. glabrescens*; TPA: *Thymus pannonicus*; TPR: *T. praecox*; TPU: *T. pulegioides*; TSE: *T. serpyllum*.

Classification of *Thymus* Chemotypes by Major Terpene Compounds

A hierarchical cluster analysis was performed on the basis of major essential oil compounds in samples belonging to seventy-three populations, resulting in nine clusters, which were primarily separated into two main groups. The first cluster included chemotypes with thymol (T) chief compounds, mainly including *T. pannonicus*; however, *T. glabrescens* and *T. pulegioides* were also represented, irrespectively of the place of origin (Figure 13). Chemotypes, where the role of sesquiterpene compounds was significant, were classified into the second-biggest cluster of the dendrogram. Non-phenolic monoterpenes and mixed chemotypes represented smaller subclasses. Germacrene D (GD) occurred in the essential oils grouped into a well-separated cluster and comprised samples belonging mainly to *T. praecox* and *T. glabrescens*. Further smaller groups could also be distinguished according to the main terpene compounds, as follows: third: *β*-caryophyllene (CAR); fourth: *p*-Cymene (CY) and thymol (T); fifth: geraniol (G); sixth: linalool (L); seventh: *β*-Cubebene (CU); and eighth: carvacrol ©. *T. serpyllum,* with its special essential oil composition (geranyl-isobutyrate: GEB), could be found in an extreme position in cluster 9 (Figure 13). Based on the classification of chemotypes shown by the dendrogram, our results on the essential oil chemotypes of *Thymus* species native to Hungary were confirmed by a cluster analysis.

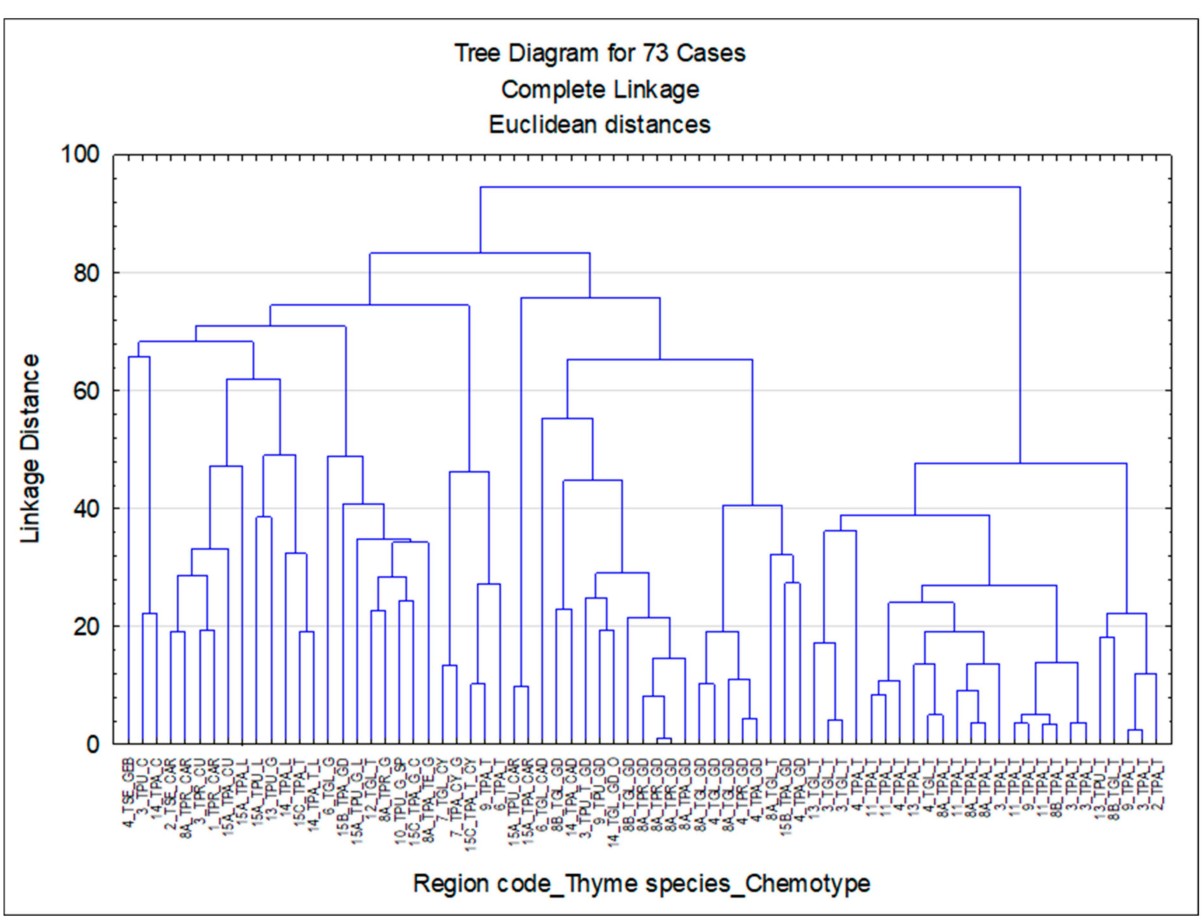

**Figure 13.** Classification of *Thymus* populations found in different regions of Hungary based on the major terpene compounds detected. Legends: Abbreviations of species names: TGL: *T. glabrescens*; TPA: *Thymus pannonicus*; TPR: *T. praecox*; TPU: *T. pulegioides*; TSE: *T. serpyllum* Region codes: see Table 1; Abbreviation of chief compounds: C: carvacrol; CAD: cadinene; CAR: *β*-caryophyllene; CU: *β*-cubebene; CY: *p*-cymene; G: geraniol; GD: germacrene-D; GEB: geranyl-isobutyrate; L: linalool; O: cis-*β*-ocymene; SP: spathulenol; T: thymol; TE: *γ*-terpinene.

## 4. Discussion

While the variability of *T. vulgaris* (garden thyme) has been extensively studied [28], less is known about the Central European wild thyme taxa. As proven in *T. vulgaris*, the chemotype patterns were a result of an adaptation process to diverse and well-defined environmental conditions [29]. According to Thompson (2002), essential oil polymorphism was considered a marker of adaptation to a particular environment. Long-term adaptation processes led to a genetically controlled essential oil biosynthesis with different monoterpenes (thymol, carvacrol, geraniol, terpineol, linalool, or thujanol) at the end of the six branches of the biosynthetic chain (regulated by the epistatic series of five loci with the following sequence: G > A > U > L > C > T) [30]. Nevertheless, no theory for the evolution of numerous further known chemotypes of taxa belonging to the sect. *Serpyllum* has been verified yet, despite the numerous data reported on the essential oil diversity.

Based on the data presented in the literature available, a considerable level of infraspecific and interspecific diversity could be predicted for the five collective species native to Hungary as well. According to our hypothesis, the drug quality of collected wild thyme samples is likely to be variable owing to the diverse habitat conditions and plant communities distributed in our country. As the occurrence of *T. serpyllum* is rather scarce, the raw material of *Serpylli herba* is to be supplied by other, widely distributed, productive taxa with high essential levels and adequate EO compositions. Consequently, the main objectives of our experiments were to explore the native *Thymus* populations in diverse regions of Hun-

gary, record taxonomical, ecological, and coenological observations, and evaluate the drug quality with respect to the essential oil content and composition. We aimed to determine the factors influencing the occurrence of different taxa and the essential oil properties as well. Valuable areas and taxa were then selected for collection, and outstanding, productive chemotypes were subjected to gene reservation and introduction into cultivation.

The identification of the species exclusively using morphological traits was a rather difficult task in natural populations. Ontogeny (flowering period), habitat features, and the type of plant community provided additional information to recognize the existing taxa. Moreover, we observed further morphological variability in dry grassland associations where populations of the same species (e.g., *T. glabrescens*, *T. pannonicus*) may appear with hairy and naked shoots or both. Among the humid conditions of mountain meadows (Mátra Hills, Zemplén Hills, etc.), the occurrence of different odors and color varieties was detected within the same population of *T. pulegioides* or *T. pannonicus*.

Concerning the frequency of occurrence of the indigenous populations of the Thymus species, we could conclude that *T. pannonicus* (Hungarian thyme) was found to have the highest frequency (38 sites), followed by *T. glabresbcens* (common thyme: 17 sites), *T. pulegioides* (mountain thyme: 9 sites), and *T. praecox* (creeping thyme: 8 sites), while *T. serpyllum* (wild thyme: 2 sites) was the least abundant among the model areas. The highest frequency of occurrence of TPA, TGL, and TPU in quite distinct habitats is likely to be related to the generalist character of the species, showing high adaptability to different circumstances.

Our results on the chemotype pattern of five *Thymus* species are partly in accordance with previous findings originating from diverse localities in the surrounding countries. In the case of *T. pannonicus,* previously available data indicate the existence of thymol, carvacrol, thymol/carvacrol, geraniol, and carveole chemovarieties from Ukraine [31], while thymol and $\alpha$-terpinyl-acetate chemotypes were reported from Bosnia [11]. Further $\alpha$-terpynyl-acetate, thymol, thymol/*p*-cymene and *p*-cymene/thymol, linalool, geraniol, and $\gamma$-muurolene/$\beta$-caryophyllene chemovarieties were noted in Slovakia [13,32]. A chemotype containing a high level of geranyl acetate in the essential oil was also described in a German population [33]. Moreover, Serbian authors have found a population in Vojvodina Province (Pannonian lowlands) with lemon-scented essential oil containing geranial and neral, and $\alpha$-pinene and germacrene-D were also detected as leading monoterpenes, where the role of the latter sesquiterpene in a chemotype pattern was emphasized for the first time [5,34]. Recently, a germacrene D/$\beta$-caryophyllene chemovariety was found in the Eastern Rodope Mountains, Bulgaria [22]. Numerous new data were found in our studies as well, where wider spectra of chief essential oil compounds have already been presented in Hungary rather than the above-mentioned sources [19,21,35]. In addition, several new chemotype patterns were detected in our studies from new locations with combinations of thymol, geraniol, geranyl acetate, *p*-cymene, and carvacrol as major monoterpenes, while sesquiterpenes were represented by germacrene D, $\beta$-caryophyllene, farnesol, caryophyllene oxide, $\beta$-farnesene, and $\delta$-cadinene. A special chemotype with a combination of mono- and sesquiterpenes was recorded in Jósvafő, Aggtelek Hills, including geraniol, geranyl acetate, and $\beta$-bisabolene as chief compounds. The wide variety in the EO composition reflects the diverse ecological conditions that exist in the mountainous areas of Hungary. Recent findings have confirmed that *T. pannonicus* has a perspective in cultivation, as promising experimental data were reported on homogenous drug quality, adequate amounts of essential oil, and stability in EO composition [36]. Our data from original habitats also supports the fact that Hungarian thyme has a high tolerance for diverse environmental conditions, which can be an advantageous trait when facing climatic changes and raising incidences of extremities.

A high diversity of major terpene compounds was also detected in the essential oils of *T. glabrescens* populations distributed widely among various habitat conditions [35]. Most of the essential oil samples contained sesquiterpenes (germacrene D, $\beta$-caryophyllene, $\tau$-cadinol or $\beta$-cadinene germacrene D, $\beta$-caryophyllene, caryophyllene oxide, etc.), while the occurrence of

monoterpenes (thymol, geraniol) was rare, such as major compounds. Our results contradict the known data in the literature, as a geraniol/neryl acetate chemotype was recorded in Romania [37], while the major essential oil compounds of Serbian and Bulgarian *T. glabrescens* populations were found to be thymol and $\gamma$-terpinene [22,38], respectively.

Concerning *Thymus praecox*, our data correspond to the volatile oil composition of *T. praecox* subsp. *skorpilii* var. *skorpilii* from Turkey, where geraniol was the principal constituent (24.2%), but, in contrast to *T. praecox* in Hungary, it was accompanied by terpinyl-acetate, geranyl-acetate, linalyl-acetate, and linalool, without any important sesquiterpenes. Other authors [30] have mentioned further essential oil compositions in which $\beta$-caryophyllene, *cis*-nerolidol, hedycaryol, germacra-1(10),5-dien-4-ol, or germacra-1(10),4-dien-6-ol occurred as sesquiterpenes. In our studies, beside geraniol in a combined chemotype [19], a general dominance of sesquiterpenes was detected, where germacrene D, $\beta$-caryophyllene, $\beta$-cubebene, and caryophyllene oxide were found to be major compounds in the essential oil samples of different origins.

The essential oil diversity of *T. pulegioides* has already been thoroughly examined as an element of the habitats of alpine or boreal regions [36,39]. The European populations of the species are predominantly described as having phenolic (thymol/carvacrol) chemotypes. On the contrary, Bulgarian authors have found a new chemotype with $\alpha$-terpinyl acetate in the Vlahina Mts., while another paper recently reported a wild mountain thyme population with $\alpha$-terpineol, geraniol, and carvacrol as major EO compounds from the Carpathians/Ukraine [22,23]. Moreover, we have also pointed out the role of sesquiterpenes in the formation of these chemotypes. In the case of lemon-scented varieties, not only did geraniol, geranial (citral B), and neral (citral A) appear in the volatile oils, but also linalool and an ester derivative, linalyl acetate, were detected. Our data were supplemented with the literature data on mountain thyme chemotypes.

The chemotype patterns revealed that the examined *Thymus serpyllum* populations were unique because the essential oil of the Bakony (Fenyőfő) was dominated by geraniol and geranyl isobutyrate, while the others in the Somogy Hills (Nagybajom) contained $\tau$-cadinol, caryophyllene oxide, and $\beta$-cubebene, respectively. In previous studies, however, thymol and related compounds were found to be major essential oil compounds that met the requirements of the respective standards [16,18]. Recently, new chemotypes with linalool as well as $\alpha$-terpineol and geraniol were also found in native populations of the Carpathians in Ukraine [23].

Further research is needed to evaluate the role of different genetic and/or environmental factors in determining chemotype patterns and their distribution in different areas. Correlations between relative percentages of several essential oil compounds and edafic conditions have already been found, but there are still open questions in this field.

## 5. Conclusions

Based on the diverse ecological conditions existing in different localities of Hungary, considerable essential oil polymorphism was found, possibly due to the outstanding adaptability of *T. pannonicus*, *T. glabrescens*, and *T. pulegioides*, respectively. Numerous chemotypes have been detected producing major volatiles of monoterpene or sesquiterpene structures. The role of sesquiterpene compounds was also proven to be important in determining the chemical character of the volatile oils, either representing independent sesquiterpenic chemovarieties or accompanying other major monoterpenes.

As the occurrence of *T. serpyllum* is rather scarce in Hungary, the raw material of *Serpylli herba* Ph. Eur. is important to be supplied by other, widely distributed, productive taxa with high essential levels and adequate EO compositions. It was established that *T. pannonicus* could be suggested for this purpose, as its essential oil content is generally high in natural habitats and, in most cases, rich in either thymol and/or its precursors (*p*-cymene/$\gamma$-terpinene), like those of *Thymus vulgaris*. The other species are either less abundant in the country (TSE), linked to special ecological conditions (TPU, TPR), or

provide substandard drug quality (TGL, TPR) with respect to the essential oil content and composition.

Further efforts suggested introducing the most valuable taxa into the horticultural systems where homogeneous drug quality with high amounts of thymol containing essential oil can be ensured. Chemotypes with special essential oil patterns are proposed to be subjected to gene reservation.

The summarized data on the occurrence, essential oil properties, and chemotype patterns of Hungarian Thymus species may serve as a supplement to the knowledge base about the chemical diversity of *Thymus* essential oils.

**Author Contributions:** Conceptualization, Z.P.; methodology, Z.P., S.T.-S., R.K. and B.G.; software, Z.P., P.R. and S.T.-S.; validation, Z.P. and S.T.-S.; formal analysis, Z.P., S.T.-S. and B.G.; investigation, Z.P., R.K., J.C. and É.N.; resources, Z.P., É.N. and J.C.; data curation, Z.P. and P.R.; writing—original draft preparation, Z.P.; writing—review and editing, Z.P. and S.T.-S.; visualization, P.R. and S.T.-S.; supervision, Z.P.; project administration, Z.P. All authors have read and agreed to the published version of the manuscript.

**Funding:** This research was funded by the National Scientific Research Fund (Hungary), grant number OTKA F 43555, the TÁMOP-4.2.1/B-09/1/KMR-2010-0005 project, and the Bolyai János Scientific Grant (Zsuzsanna Pluhár) of the Hungarian Academy of Sciences, respectively.

**Data Availability Statement:** Data are contained within the article.

**Acknowledgments:** We would like to thank Hella Baráthné Simkó, a former Ph.D. student, for their kind assistance with the collection and processing of data, and further graduate students, in particular Emese Szabó, Adrienn Pintér, and Balázs Marton, who were involved in *Thymus* research. We also highly appreciate the continuous support of Klára Ruttner in laboratory analysis and the technical support of Péter Rajhárt and Ferenc Erdei at the experimental station of the university.

**Conflicts of Interest:** The authors declare no conflicts of interest.

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
