# Peer review of "Variations in Essential Oil Composition and Chemotype Patterns of Wild Thyme (Thymus) Species in the Natural Habitats of Hungary"

_horticulturae, doi:10.3390/horticulturae10020150_

Round 1

Reviewer 1 Report

Comments and Suggestions for Authors

Dear Editors, Dear Authors,

Presented manuscript deals with phytochemical compostition and chemotypes of Thymus species collected from several localities in Hungary. Although phytochemical profiling of wild plants is important for their horticultural performance, the presented MS has several downsides that should be addressed so to be publishable in Horticulturae:

1. Introduction must be updated. There is new data that should be considered, including in Horticulturae (see https://www.mdpi.com/2304-8158/5/4/70; https://www.mdpi.com/2311-7524/8/12/1218; https://link.springer.com/article/10.1007/s42452-020-2938-2)

2. M&M and Results: Studied localities are presented only by the type of base rock and soil type (needs improvement as well, See file with track for further remarks and comments.) that is not sufficient description given biosynthetic profiles are influenced by number of factors. However, data in tables and figures is given grouped by regions that include several types of rock/soil (e.g. 3 Balaton Uplands) and it is very hard to follow the 5 species growing in various combinations throughout the text. As in some localities the species coexist it is also questionable why the Authors are reporting "frequency of occurrence" of "populations" without any information on the specific abundance or surface area, especially when commenting harvesting in the Results section. Additionally, it remained unclear when the samples were collected/studied as yields and phytochemical compostition vary seasonally and between years - "with 3 to 6 replicates were collected per site and essential oil data of 2 subsequent years were also evaluated in order to describe the true chemotypes of the localities" is stated in the M&M but there are no results presented further in the text. Chemotype patterns are presented once in figures 4-8, then in tables 3 and 4 and again repeated in the text that makes the reading extremely troublesome. It is not clear what are Fig. 9 and 10 - some variant of Fig. 4?

3. Discussion starts around the figures 9-11, prepared by mixing the data for all species, kind of T. serpyllum L. s.l. approach. What is the idea in designing new chemotypes based on low yielding populations when the pharmacopeic standards are targeted? Are growing conditions the main contributing factor(s) or genetics? Are these populations substantial in area for prospective harvesting? Are they currently exploited? In general the section is just continuation of the results in a different form and the discussion is only between lines 630-682. The focus is placed on the chemotype patterns but not on the (possible) yields.

4. Conclusions are not based on the results and include mostly generalizations.

Reviewer 2 Report

Comments and Suggestions for Authors

ADDITIONAL COMMENTS

“Phytochemical composition and chemotype pattern distribution of wild thyme (Thymus) species in natural habitats of Hungary” 

In this manuscript, the authors investigated the essential oil of Thymus spp.

The manuscript provided to me for review is interesting. In recent years, secondary metabolites such as components of essential oils have been of great interest. They have proven important biological properties as well as a beneficial effect on human health.

The present manuscript authors identify the chemical composition of the essential oil of various Thymus spp.

The present work is interesting, detailed, and thorough research has been carried out. For this, I give my consent for publication in your magazine.

I noticed an inaccuracy that bothered me:

Figure 3. „Essential oil content (mL/100 g DW, mean±SD) of samples originating from Thymus spp. populations surveyed in natural habitats belonging to different regions in Hungary“.

Ø  The authors determined the yield of EO from Thymus spp. The statistics are disturbing. In some places SD>mean, which means that RSD is 100 and over 100%, in others it is over 80%. The way they extract EO varies widely. Can the authors explain why this is the case?  You use a Clevenger apparatus to isolate EO. Could it be that hydro-distillation does not give good reproducibility?

Ø  In the Statistical analysis section, the authors did not indicate how many times the experiments were conducted, i.e. how many repetitions were made to statistically process the results.

Ø  The subject of the article I found an error

            Line 2: compostition ------ > composition

Reviewer 3 Report

Comments and Suggestions for Authors

The manuscript by Zsuzsanna Pluhár et al. describes the chemical characterization of a population of Thymus species collected in Hungary. Although this manuscript has interesting and relevant information for readers and the SI, several issues were detected during peer review that should be addressed before further consideration.

1.      Title: It can be improved. In my humble opinion, the two first terms in the title can be merged, retaining only "chemotype pattern distribution" and removing "phytochemical composition" since the chemotype recognition requires chemical characterization. Therefore, I recommend this more concise title: "Chemotype pattern distribution of wild thyme (Thymus) species in Hungarian natural habitats.

2.      The format and presentation of this manuscript version do not strictly follow the journal template.

3.      Line 17-19: This sentence should be improved, providing a better introductive passage since it is missing. Indeed, if this sentence is modified, the second sentence (lines 19-23) should also be rewritten.

4.      Line 41: The botanical family and subfamily do not require italics.

5.      Line 82-95: This passage must be improved since it is unclear. For instance, the aim and scope of the study are not well stated.

6.      Lines 139 and 147: Why were the temperature programs of GC/FID and GC/MS and conditions not identically performed? Explain this issue.

7.      Line 143: Why was ester percentage if terpenoids were mainly found?

8.      Line 146: The column used for GC/MS is not informed.

9.      Figure 3: Considering that the data collected in this manuscript comes from wild plants, these figures must be box plots instead of bar plots. The box plots can better orient the data distribution and better visualize the essential oil content per Thymus species and location.

10.   Line 257: What does "chief compound" mean? Is this related to the main compound? Revise and be consistent throughout the manuscript.

11.   The discussion section is highly descriptive. Other passages summarize the results or involve introductive ideas (e.g., lines 462-472 comprise introduction), and other exhibited data analysis (i.e., Figures 9-13), which must be part of the results and must be accordingly moved. In fact, this section seems to be a results section rather than a discussion section, and they (results and discussion) must be reorganized accordingly. Hence, the discussion can exploit the results to provide a more comprehensive discussion, e.g., under comparison with previous studies or theory. In this regard, a subheading division is recommended as a good strategy to organize the discussion better.

12.   The conclusion section also summarizes the results. After analyzing the results from the mechanistic point of view, it is recommended that conclusions be rewritten into conceptual findings.

Comments on the Quality of English Language

The main limitation of this manuscript is the language quality and presentation since several grammar and stylistic issues are found throughout it, making it challenging to follow the information and findings. A language editing service is recommended.

Reviewer 4 Report

Comments and Suggestions for Authors

Based on the provided text, the article "Phytochemical composition and chemotype pattern distribution of wild thyme (Thymus) species in natural habitats of Hungary" appears to be a comprehensive study on the diversity and characteristics of wild thyme species in Hungary. Here's a brief analysis of its content and suitability for publication:

Critical Points: I believe the title could be enhanced with the term "volatile phytochemicals" or "essential oils".

Suggestion: Volatile Phytochemical Composition and Distribution of Chemotypic Patterns of Wild Thyme Species (Thymus) in Natural Habitats of Hungary

or

Variation in Essential Oil Composition and Chemotype Patterns of Wild Thyme Species (Thymus) in the Natural Habitats of Hungary

Abstract: The abstract should begin with a concise and clear introduction to the subject matter. Instead of directly stating "data were summarized," it would be more engaging to briefly introduce the relevance of studying Thymus species in Hungary. Or Present the proposed objective is.... Group related information to improve flow and comprehension. For instance, details about the Thymus species, their habitats, and their occurrence can be presented together, followed by information on essential oil content and chemotype patterns. Clearly state the implications of your findings, particularly the potential of T. pannonicus for cultivation and its adaptability to varying environmental conditions. This helps readers understand the significance of your work.

"habitat preferencies" should be "habitat preferences". "Preferencies" is not a word in English; the correct term is "preferences."

"was shown for T. pannonicus" should be "were shown for T. pannonicus". The subject "average amounts" is plural, so the verb should be "were" instead of "was."

"low EO accumulating ability was detected at T. glabrescens" should be "low EO accumulating ability was detected in T. glabrescens". When referring to characteristics found within a species, "in" is the appropriate preposition, not "at."

"the frequency of thymol chemotype was the most considerable" could be better phrased as "the thymol chemotype was the most frequent". The term "considerable" is typically used to describe size or amount, not frequency.

"numerous further monoterpene" should be "numerous other monoterpenes". "Further" is less specific than "other," and "monoterpene" should be plural to match "numerous."

"dominated as well as combined" is a bit awkward. A clearer phrasing could be "were dominant, as well as combined".

"has perspective in cultivation with homogenous drug quality" is unclear. It could be rephrased to "shows potential for cultivation with uniform drug quality". "Perspective" is not the right word here; "potential" is more appropriate.

"facing to the climatic change and extremities" should be "facing climate change and extremities". The preposition "to" is not necessary after "facing," and "the" before "climate change" is not needed.

Keywords: The words included cannot be in the title. Choose other words for here.

Introduction

The introduction is well-structured, starting with a general overview and narrowing down to the specifics of your study. However, some sentences are quite long and could be split for better readability. 

You've provided a thorough background, including the history of use in traditional medicine and recent scientific findings. This sets a strong foundation for understanding the context of your study.

- Clearly state the research problem at the beginning.

- Bring the statement of objectives forward and make it more prominent.

 - Consider breaking longer sentences into shorter ones for clarity.

- Ensure consistent formatting, especially in the species names (italicization) and reference numbers.

Metodology

Study Area Selection: The choice of study areas based on pre-recorded floristic and coenological data may lead to selection bias, as these data may not adequately represent the diversity of thyme populations.

Species Identification: Identification based solely on literature and field data may not be sufficient for precise identification, especially for microtaxa.

Sample Collection: The collection method (3 to 6 replicates per site) may not be sufficient to represent the genetic and chemical diversity of the populations. Moreover, collecting only over two consecutive years may not capture the appropriate seasonal or annual variation.

Missing Information:

Details on Coenological Data Collection: There are no detailed information on how the coenological data were collected and analyzed.

Criteria for Population Selection: The specific criteria for the choice of the 74 populations among the study areas are not mentioned.

Chemical Analysis Methodology: There is a lack of details on how the essential oil samples were analyzed and how the chemotypes were determined.

Limitations:

Temporal Sampling: The research was conducted over a 20-year period, but it is unclear if the sampling was consistent throughout this time.

Habitat Diversity: Despite the mention of a wide range of habitats, it is unclear if all relevant habitat types were included.

Statistical Analysis: The use of ANOVA and cluster analysis is suitable, but may not be sufficient to capture the ecological and genetic complexity of thyme populations. I suggest the use of Principal Component Analysis and/or Correspondence Analysis to improve the proposed approach. If possible, it is not a requirement.

Generalization of Results: The conclusions may not be applicable to areas outside of Hungary due to specific environmental and climatic conditions.

Representativeness of New Populations: It is unclear how the 15 new locations compare with the existing populations in terms of diversity and representativeness.

Results

- In all text, use the IUPAC rule for symbols. Put p, O, E-, cis- and tau of the compounds in italics.

- Table 1 and Table 2: The summarization of the table is well defined.

- Figure 3: It would be necessary to apply a statistical test to verify the significance and differences between the studied groups. The image contributes little to the study.

- Figures 4, 5, 6, and 7: Standardize the forms that the compounds present in the table. Suggestion that the samples..

- Table 4: LRI – Use the first four decimal places, without any values after the comma. For example: 1000

- Throughout the entire text, the term 'Monoterpene hydrocarbons and Sesquiterpene hydrocarbons' should be replaced with 'Non-oxygenated monoterpenes and Non-oxygenated sesquiterpenes'. The majority of terpenoids predominantly have hydrocarbon functional groups. (See Table 4).

Table 4: All compounds were identified based on Mass and retention index. Were the calculations of the retention indices made on a homologous series of alkanes? Include this in the table and methodology. Was the quantification based on GC (Gas Chromatography)? Is the order of the substances that of elution in the column?

Figure 13: O termo: “Distância euclidiana” deve ser apresentada no local de “Linkagen Distance”.

Species and Location Diversity: The study covers a wide range of Thymus species and their habitats, indicating rich biological diversity. However, there is a disproportionate emphasis on certain species such as Thymus pannonicus and Thymus glabrescens, while others, like Thymus serpyllum, are less explored. This could lead to an incomplete understanding of Thymus biodiversity in the region.

Variation in Occurrence Frequency: The data shows significant variations in the frequency of occurrence of the species, with T. pannonicus being the most common. The accuracy of these frequencies is not clearly established and could be influenced by factors such as sampling efforts or seasonal variations.

Ecological Preferences and Geographic Distribution: The study details the ecological preferences and geographic distribution of each species, providing valuable information about their ecosystem. However, the relationship between ecological characteristics and geographic distribution could be better explored to understand how environmental factors influence the presence of each species.

Essential Oil Content: The results show significant variations in essential oil content among species and within a species at different locations. While this provides important insights into the quality and commercial potential of these plants, the connection between the chemical composition of the essential oil, ecological conditions, and practical implications for the use of these plants could be further deepened.

Essential Oil Chemotypes: The classification into different chemotypes is a crucial part of the study, but the methodology for determining these chemotypes is not clearly explained. Additionally, the relationship between chemotypes, ecology, and potential uses of the plants is an area

that could be further explored. Understanding how chemotypes vary in different environmental conditions and the impact of these variations on the therapeutic or commercial potential of Thymus essential oils could offer valuable insights.

Consistency and Comparability of Data: The study covers a long period (2000-2020) and a wide geographical area. This raises questions about the consistency of data collection and analysis methods over time and in different locations. Methodological variations could affect the comparability of the results.

Implications for Conservation: While the study provides detailed information on the distribution and ecology of Thymus species, the implications of these findings for conservation strategies are less emphasized. Greater emphasis on the practical implications of these findings for the conservation of these species and their habitats would be beneficial.

Use of Secondary Data and References: The study appears to rely significantly on secondary data and literature references. The integration and critical analysis of these sources are crucial to ensuring the accuracy and relevance of the presented results.

Contextualization with Related Studies: Although the study offers a comprehensive overview of Thymus species in Hungary, a more direct comparison with similar studies in other regions could provide a broader perspective on the diversity and ecology of these plants in a global context.

Discussion:

Some results are presented in the discussion section, which should be transferred. I believe it is not appropriate to include them there. Please italicize the 'p' values in the text.

Comments on the Quality of English Language

To avoid extending further, please perform a valuable review of the English throughout the entire text.

Round 2

Reviewer 1 Report

Comments and Suggestions for Authors

Dear Editors,

Dear Authors,

I'm pleased with the proposed changes and commend the Authors for their efforts. I only wish to point that some expressions form the Conclusions have to be reformulated :

Recent findings confirmed that T. pannonicus has perspective in cultiva- 764
tion, as promising experimental data were reported on homogenous drug quality, adequate amount of essential oil as well as stability in EO composition [38 40, 39 41]. Our data from original habitats also support that Hungarian thyme has high tolerance among diverse environmental conditions, which can be an advantageous trait when facing to the climatic changes and raising incidences of extremities.
- Such comments should be part of the Discussion.

Based on own experiences, new studies have been started recently, which are aimed to determine the tolerance limits of T. pannonicus subjected to different abiotic stress factors. - This statement together with the final sentence are rather unclear.

Comments on the Quality of English Language

Language and style should be re-checked,  after addition of new texts there are some unclear expressions in the results and conclusions.

Reviewer 3 Report

Comments and Suggestions for Authors

The authors have satisfactorily addressed my comments. However, there are some format concerns that could be improved in subsequent steps. Therefore, the manuscript can be considered for further review.

Comments on the Quality of English Language

There are still certain language, grammar, and style issues that require meticulous revision throughout the manuscript.
